# Transglutaminase 2 mediates hypoxia-induced selective mRNA translation via polyamination of 4EBPs

Sung-Yup Cho[1,2,3], Seungun Lee[4], Jeonghun Yeom[5], Hyo-Jun Kim[1], Jin-Haeng Lee[1], Ji-Woong Shin[1], Mee-ae Kwon[1], Ki Baek Lee[1], Eui Man Jeong[1,6], Hee Sung Ahn[5], Dong-Myung Shin[4], Kyunggon Kim[4,5], In-Gyu Kim[1,6]

Hypoxia selectively enhances mRNA translation despite suppressed mammalian target of rapamycin complex 1 activity, contributing to gene expression reprogramming that promotes metastasis and survival of cancer cells. Little is known about how this paradoxical control of translation occurs. Here, we report a new pathway that links hypoxia to selective mRNA translation. Transglutaminase 2 (TG2) is a hypoxia-inducible factor 1–inducible enzyme that alters the activity of substrate proteins by polyamination or crosslinking. Under hypoxic conditions, TG2 polyaminated eukaryotic translation initiation factor 4E (eIF4E)-bound eukaryotic translation initiation factor 4E-binding proteins (4EBPs) at conserved glutamine residues. 4EBP1 polyamination enhances binding affinity for Raptor, thereby increasing phosphorylation of 4EBP1 and cap-dependent translation. Proteomic analyses of newly synthesized proteins in hypoxic cells revealed that TG2 activity preferentially enhanced the translation of a subset of mRNA containing G/C-rich 5′UTRs but not upstream ORF or terminal oligopyrimidine motifs. These results indicate that TG2 is a critical regulator in hypoxia-induced selective mRNA translation and provide a promising molecular target for the treatment of cancers.

## Introduction

Inadequate oxygen availability, termed hypoxia, is a common stress that cancer cells encounter during tumor progression. Under hypoxic conditions, inhibition of translation is a critical response for tumor cell survival because translation is an ATP-consuming process that requires approximately one-third of all cellular ATP (Fahling, 2009; Chee et al, 2019). Translational control of gene expression is a convenient regulatory point because of its rapidity, requiring no transcriptional lag, which permits the cell to process new mRNA transcripts to adapt to a hypoxic environment (Spriggs et al, 2010; Chee et al, 2019). Indeed, hypoxic stress alters a number of tumor cellular behaviors, such as metabolic reprogramming, enhanced angiogenesis, migration, and apoptotic resistance (Majmundar et al, 2010), suggesting the existence of a mechanism that controls the selective translation of mRNAs responsible for these phenotypes.

Eukaryotic translation initiation factor 4E (eIF4E) is a binding protein to the 5′ cap structure of mRNA and, together with scaffolding protein eIF4G and RNA helicase eIF4A, forms the eIF4F complex. After binding of eIF4E to the cap, the eIF4F complex triggers eIF4A-mediated mRNA unwinding and eIF4G-induced recruitment of the 43S preinitiation complex that leads to the scanning of the start (AUG) codon and ribosomal 60S subunit joining (Siddiqui & Sonenberg, 2015). The formation of the eIF4F complex is, thus, critical for cap-dependent translation initiation. This rate-limiting step is regulated by interactions between eIF4E and eIF4E-binding proteins (4EBP1, -2, and -3). Binding of 4EBPs to eIF4E inhibits eIF4F complex formation by interfering with the eIF4E–eIF4G association. Mammalian target of rapamycin complex 1 (mTORC1) phosphorylates 4EBPs, thereby dissociating eIF4E and promoting eIF4F complex assembly (Siddiqui & Sonenberg, 2015). Various stimuli modulate mRNA translation through the regulation of mTORC1 activity. Growth factors activate mTORC1 through phosphatidylinositol-3-kinase (PI3K)–AKT and MAPK signaling pathways, whereas hypoxic stress inhibits mTORC1 via adenosine monophosphate–activated protein kinase (AMPK) signaling and by the hypoxia-inducible factor 1 (HIF1)–induced REDD1 (regulated in development and DNA damage response-1) and BNIP (BCL2/adenovirus E1B 19 kd-interacting protein) (Sengupta et al, 2010; Roux & Topisirovic, 2018). Despite suppressed mTORC1 activity under hypoxic conditions, mRNA translation of genes for cellular survival and metabolic reprogramming is enhanced in tumor cells (Spriggs et al, 2010). However, the detailed molecular mechanisms underlying the regulation of 4EBP phosphorylation in response to hypoxic stress are not fully understood.

Transglutaminase 2 (TG2) is a calcium-dependent enzyme that modifies proteins by catalyzing acyl transfer reaction between

---

[1]Department of Biochemistry and Molecular Biology, Seoul National University College of Medicine, Seoul, Korea  [2]Department of Biomedical Sciences, Seoul National University College of Medicine, Seoul, Korea  [3]Cancer Research Institute, Seoul National University College of Medicine, Seoul, Korea  [4]Department of Biomedical Sciences, University of Ulsan College of Medicine, Seoul, Korea  [5]Department of Convergence Medicine, Asan Medical Center, Seoul, Korea  [6]Institute of Human-Environment Interface Biology, Seoul National University College of Medicine, Seoul, Korea

Correspondence: csybio@snu.ac.kr; igkim@plaza.snu.ac.kr; kimkyunggon@gmail.com

protein-bound glutamine residue and protein-bound lysine residue (cross-linkage) or polyamine (polyamination) (Tatsukawa et al, 2016). TG2-mediated modifications modulate the activity of substrate proteins, such as inhibitor of NF-κB (IkB) or caspase 3, which trigger inflammation or promote chemoresistance (Lee et al, 2004; Jang et al, 2010). Intracellular TG2 is enzymatically dormant under physiological conditions, partly because of low intracellular calcium concentration (Jeon et al, 2003b; Siegel & Khosla, 2007), but is activated by stressors such as $H_2O_2$ (Jeong et al, 2009), chemotherapeutic agents (Cho et al, 2012), and endoplasmic reticulum stress (Kojima et al, 2010; Kuo et al, 2011; Lee et al, 2014). TG2 is also involved in tumor cellular function: tumor cells under hypoxic stress exhibit an HIF1-dependent increase in TG2 expression and intracellular activity (Jang et al, 2010), and, in a mouse xenograft model, TG2-depleted tumor cells show reduced cell growth and viability (Jang et al, 2010). TG2 plays roles in cellular responses to hypoxia by regulating transcription (such as NF-κB) and posttranslational modification (such as caspase 3) (Jang et al, 2010). However, the impact of TG2 on translation reprogramming in hypoxia has been little investigated.

In this study, we identified 4EBPs as TG2 substrates under hypoxic conditions. TG2-mediated 4EBP1 polyamination increased its binding to mTORC1, the quantity of phosphorylated 4EBP1, and cap-dependent translation, resulting in enhanced translation of selective mRNAs involved in cell–cell interaction, macroautophagy, and RNA metabolism. These results define a new mechanism by which hypoxia regulates the cap-dependent translation of selected mRNAs and provide a candidate target for cancer therapeutics.

# Results

### TG2 polyaminates 4EBP1 in hypoxic cells

To investigate the role of TG2 in hypoxic translational control, we searched the target substrates of TG2 among translation-related proteins. A549 cells, which showed the enhanced TG2 activity in response to hypoxia (Fig S1A), were cultured with a polyamine analog, 5-biotinamidopentylamine (BP), and BP-incorporated proteins due to TG2 activity was detected by streptavidin pull-down and Western blotting (Fig S1B). Among several translation-related proteins, we found that 4EBP1 is polyaminated in a TG2-dependent manner in cells under hypoxic conditions (Fig 1A). The pulled-down 4EBP1 corresponded to lower band of 4EBP1 in the input (Fig 1A). Using purified TG2 and 4EBP1, an in vitro solid-phase microtiter plate assay showed hyperbolic kinetics in the substrate–velocity curve; the calculated $K_m$ for BP was 35.8 $\mu$M, confirming that 4EBP1 is a substrate of TG2 (Fig 1B). Previous reports show that several substrates are cross-linked and/or polyaminated by TG2, producing insoluble aggregates such as crystalline, IkB, and caspase 3 (Shridas et al, 2001; Lee et al, 2004; Jang et al, 2010). We determined the type of 4EBP1 modification via Western blot. In the reaction with TG2, only 4EBP1 incorporating BP was detected (Fig 1C); no homo-dimer, tetramer, or insoluble aggregate was present (Fig 1D). Moreover, polyaminated 4EBP1 exhibited no alteration in solubility (Fig S1C).

Putrescine, spermidine, and spermine are polyamines with different numbers of amino groups that are found in high concentrations in eukaryotic cells. We compared the affinity of each polyamine for TG2 using a competitive incorporation assay. When added to the reaction mixture, spermidine showed higher inhibitory activity for BP incorporation by 4EBP1 than spermine or putrescine (Fig 1E), indicating that 4EBP1 might be spermidinylated by TG2 in the cells. In addition, because polyamines are polycationic compounds, we tested whether TG2-mediated polyamination of 4EBP1 could be affected by its phosphorylation status. Cell lysates were treated with phosphatase, and BP incorporation by 4EBP1 was assessed. 4EBP1 polyamination was greater in phosphatase-treated lysates than untreated controls (Fig 1F). Enhanced polyamination of non-phosphorylated 4EBP1 was also observed in the cells treated with PP242, an mTORC1 inhibitor (Fig 1G). Moreover, when expressed in cells, the phosphorylation-defective mutant (T37,46A) exhibited augmented BP incorporation relative to wild-type 4EBP1 in response to hypoxia (Fig 1H), confirming that non-phosphorylated 4EBP1 is preferentially polyaminated by TG2 in hypoxia.

### Polyamination sites are conserved in 4EBP1, -2, and -3

In the TG2-catalyzed transamidation reaction, a glutaminyl residue of the substrate protein acts as a polyamine acceptor (Tatsukawa et al, 2016). To investigate the functional significance of 4EBP1 polyamination, we identified glutamine residues of 4EBP1 polyaminated by TG2. Because no consensus polyamination sites for TG2 are known, each of the four glutamine residues in 4EBP1 was mutated to alanine, and the level of BP incorporation of the mutants was assessed in A549 cells in vitro. Among the mutant derivatives, the Q28A, Q113A, and Q95A mutants of 4EBP1 were poorly polyaminated by TG2 relative to wild-type 4EBP1. Moreover, Q28,95,113A triple-mutant 4EBP1 exhibited complete loss of BP incorporation (Fig 2A). Mass spectrometric analysis confirmed the spermidinylated Gln28 and Gln113 residues of wild-type 4EBP1 (Figs 2B and S2A), indicating that Gln28 and Gln113 are major polyamination sites.

Comparison of amino acid sequences showed that Gln28 and Gln113 are conserved in all human 4EBPs and across 4EBP1 in various species (Fig S2B). Based on conserved polyamination sites among 4EBPs, we tested whether 4EBP2 and 3 are substrates of TG2. As with 4EBP1, solid-phase microtiter plate assay (Fig 2C) and Western blot analysis (Fig 2D) demonstrated that 4EBP2 and 3 were also polyaminated by TG2 and have lower $K_m$ values for BP than did 4EBP1 (20.7 and 20.3 $\mu$M for 4EBP2 and 4EBP3, respectively). In addition, the conserved glutamine residues in 4EBP2 (Q29 and Q115) and 4EBP3 (Q95) were crucial in TG2-mediated polyamination (Fig 2E).

### TG2 preferentially polyaminates eIF4E-bound 4EBP1 in hypoxic cells

Non-phosphorylated 4EBP1 binds to eIF4E and is a preferred substrate for TG2, as shown above. To demonstrate that polyamination of 4EBP1 may influence its ability to associate with eIF4E, we compared the level of polyamination between free and eIF4E-bound 4EBP1. To this end, phosphorylation- and eIF4E-binding–defective 4EBP1 (T37,46A/LM59,60AA) (Yanagiya et al, 2012) was expressed in cells and BP incorporation by 4EBP1 was evaluated after exposure to hypoxia. Unexpectedly, this mutant exhibited a

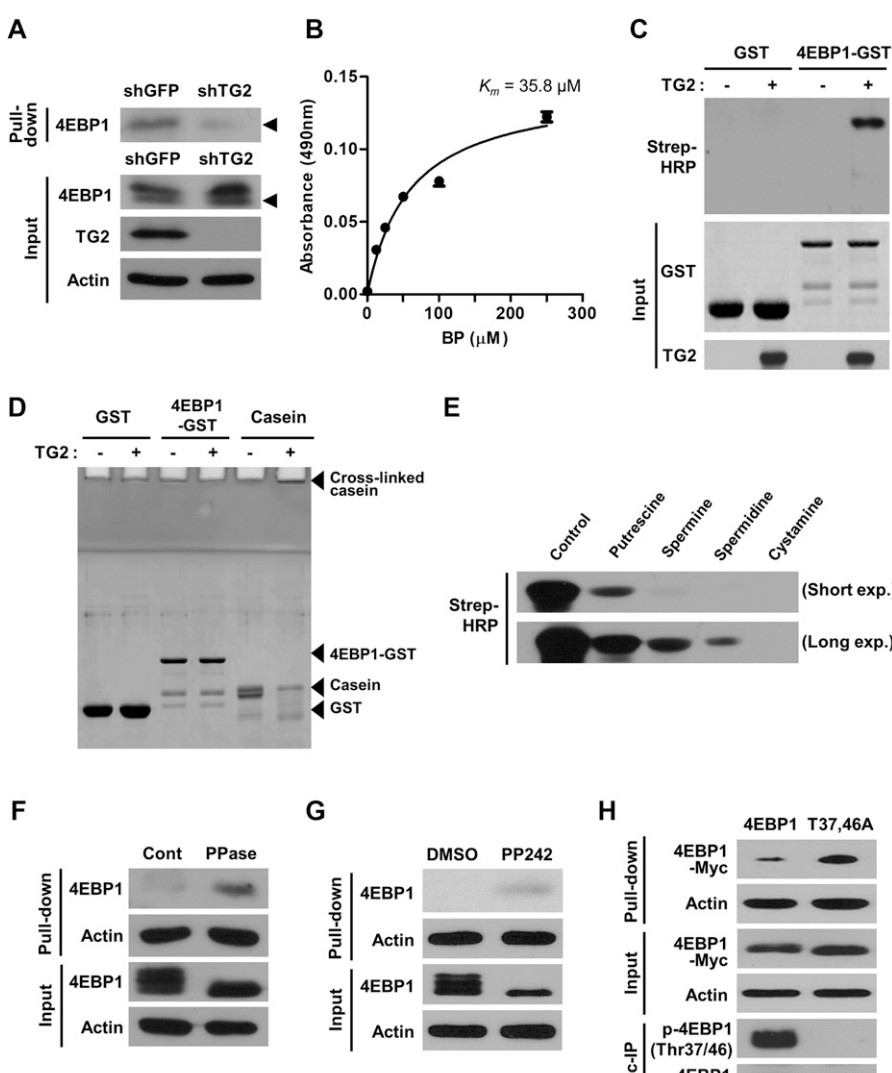

**Figure 1. TG2 polyaminates 4EBP1 in hypoxic cells.**
**(A)** In situ incorporation of 5-BP in hypoxia. Wild-type (shRNA for GFP [shGFP]) and TG2-knockdown (shRNA for TG2 [shTG2]) A549 cells were first exposed to hypoxia (1% O2) for 6 h in the presence of BP, and after that BP-incorporated 4EBP1 was detected by streptavidin pull-down and Western blot analysis. Triangles indicate the 4EBP1 of same molecular weight. **(B, C)** In vitro TG2-mediated modification of 4EBP1. 4EBP1-GST was incubated with BP and TG2, and BP incorporated into 4EBP1 was detected by solid-phase microtiter plate assay (B) or Western blot analysis (C). A range of BP concentrations (0–250 $\mu$M) were used to determine $K_m$. **(C)** The data represent the mean values ± standard deviations based on experiments that were performed in triplicate (C). **(D)** In vitro TG2-mediated crosslinking of substrates. 4EBP1-GST, GST, and casein were incubated with TG2, and cross-linked proteins were evaluated by Coomassie blue staining. **(E)** Competitive TG2-mediated polyamine incorporation of BP into 4EBP1. 4EBP1-GST was incubated with TG2 and BP (100 $\mu$M) in the presence of polyamines (putrescine, spermine and spermidine; 1 mM). Cystamine is a TG2 inhibitor, which was used for negative control. Upper and lower panels showed the result from short and long exposure (exp.) times, respectively. **(F, G, H)** Dephosphorylation of 4EBP1 was required for TG2-mediated polyamination. A549 cell lysates were incubated with BP (200 $\mu$M for 1 h) after 4 h treatment of calf intestinal alkaline phosphatase (F). **(G, H)** A549 cells were treated with mTORC1 inhibitor, PP242 (1 $\mu$M) for 1 h (G) or transfected with phosphorylation-defective mutant 4EBP1 (T37,46A; H), and then were exposed to hypoxia (1% O$_2$) with BP (200 $\mu$M) for 6 h. BP-incorporated 4EBP1 proteins was detected by streptavidin pull-down and Western blot analysis. **(H)** The phosphorylation status of Myc-immunoprecipitated (Myc-IP) 4EBP1 was evaluated by Western blot analysis (H). Cont, control; Myc, Myc tag; PPase, alkaline phosphatase.

greatly reduced level of polyamination relative to the phosphorylation-defective mutant (Fig 3A). A similar reduction in 4EBP1 polyamination was observed in eIF4E-depleted cells (Fig 3B), indicating that binding of 4EBP1 to eIF4E is required for 4EBP1 polyamination by TG2. Although knockdown of eIF4E slightly increased the phosphorylated 4EBP1, the decrease of 4EBP1 polyamination was significant compared with the increase of 4EBP1 phosphorylation (Fig 3B).

To test whether eIF4E-4EBP1 complex acts as a substrate, we examined the interaction between eIF4E and TG2. When cell lysates were incubated with anti-TG2 antibody, eIF4E co-immunoprecipitated (Fig 3C). Conversely, when cell lysates were incubated with eIF4E-GST, TG2 was detected in the pull-down GST–sepharose protein complexes (Fig 3D). The direct interaction of eIF4E with TG2 was further confirmed by in vitro 7-methyl GTP (mRNA cap structure) affinity chromatography using purified eIF4E and TG2 (Fig 3E). Moreover, interaction domain mapping revealed that eIF4E binds to the catalytic core domain of TG2 (Fig 3F), whereas 4EBP1 interacts with the barrel 1 domain of TG2 (Fig 3G). We then compared the ability of wild-type and polyamination-defective

4EBP1 (4EBP1$^{Q28,95,113A}$) to interact with eIF4E and TG2. Both wild-type and mutant 4EBP1 were co–pulled-down with TG2 and eIF4E but showed no significant difference in binding affinity for TG2 and eIF4E (Fig 3H). Interestingly, despite direct binding between eIF4E and TG2, poly-amination of eIF4E was not observed in the BP incorporation assay among cells exposed to hypoxia (Fig 3I). Together, these results demonstrate that TG2 preferentially polyaminates eIF4E-bound 4EBP1 in hypoxic cells.

## 4EBP1 polyamination increases its phosphorylation by enhancing binding affinity for Raptor

Phosphorylation of 4EBP1 induces a release of eIF4E due to a lowered eIF4E affinity, thereby initiating mRNA translation (Siddiqui & Sonenberg, 2015). To test whether polyamination of eIF4E-bound 4EBP1 may influence phosphorylation of 4EBP1, we compared 4EBP1 phosphorylation in wild-type and TG2-depleted A549 cells cultured under hypoxia. Levels of phosphorylation at Thr37 and

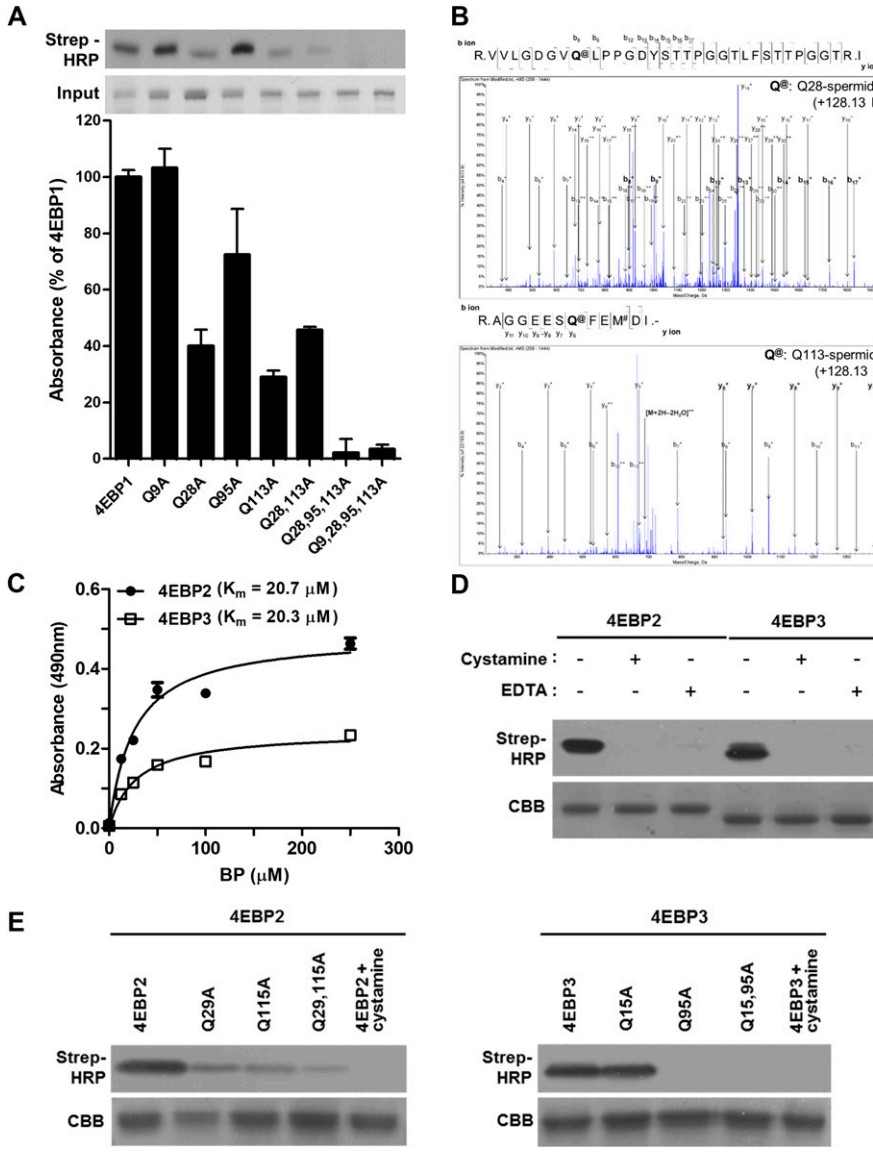

**Figure 2. Polyamination sites are conserved in 4EBP1, -2, and -3.**
**(A)** Effect of glutamine-to-alanine mutation in 4EBP1 on TG2-mediated BP incorporation. Wild-type and mutant 4EBP1-GST were incubated with BP (100 $\mu$M) and TG2, and BP-incorporating proteins were detected by Western blot analysis (upper panel) and solid-phase microtiter plate assay (lower panel). The data represent the mean values ± standard deviations based on experiments that were performed in triplicate. **(B)** Determination of TG2-catalyzed polyamination sites in 4EBP1 by mass spectrometry. Peptides containing spermidine-incorporating Q28 and Q113 are shown at the upper and lower panels, respectively. **(C, D)** 4EBP2 and 4EBP3 are substrates of TG2. **(C, D)** 4EBP2-GST and 4EBP3-GST were incubated with BP and TG2, and BP incorporated into 4EBP2 and 4EBP3 was detected by solid-phase microtiter plate assay (C) or Western blot analysis (D). A range of BP concentrations (0–250 $\mu$M) was used to determine $K_m$. **(C)** The data represent the mean values ± standard deviations based on experiments that were performed in triplicate (C). **(E)** Effect of glutamine-to-alanine mutation in 4EBP2 and 4EBP3 on TG2-mediated BP incorporation. Wild-type and mutants of 4EBP2-GST and 4EBP3-GST proteins were incubated with TG2 and BP (100 $\mu$M), and BP incorporated into proteins was detected by Western blot analysis. Samples treated with cystamine (500 $\mu$M) and EDTA (500 $\mu$M) were used as negative controls. CBB, Coomassie Brilliant Blue.

Thr46 (T37/T46) of 4EBP1 were comparable in both cells and abruptly decreased after 48 h, more significantly in TG2-knockdown cells (Fig 4A). In addition, reduced phosphorylation at Ser65 and Thr70 (S65/T70) was observed after 24 h in TG2-knockdown cells (Fig 4A). Under the same experimental conditions, no significant changes in the levels of 4EBP1, eIF4E, and p-eIF4E were observed (Fig 4A). Moreover, when eIF4E pull-down was performed with 7-methyl GTP sepharose beads, eIF4E-bound 4EBP1 in TG2 knockdowned cells increased in hypoxic conditions (Figs 4B and S3), indicating that TG2 is required for phosphorylation of 4EBP1 in hypoxia and, thus, for dissociation of eIF4E from 4EBP1.

mTORC1 phosphorylates 4EBP1 sequentially at T37/T46 and then S65/T70, but the latter responds more sensitively to growth signals or energy status (Gingras et al, 2001). To explain the decreased phosphorylation of 4EBP1 in TG2-depleted cells under hypoxia, we assessed activities of upstream kinases by measuring phosphorylation levels of Thr308 and Ser473 of AKT and Ser2448 of mTOR to assess AKT activity

and phosphorylation levels of Ser2481 of mTOR (Peterson et al, 2000) and Thr389 of S6K to assess mTORC1 activity (Sengupta et al, 2010). The overall levels of phosphorylation were decreased by hypoxia, but little significant differences in the activity of AKT and mTORC1 between wild-type and TG2-knockdown cells were observed (Fig 4C), indicating that TG2 has little effect on the activities of AKT and mTORC1. Because 4EBP1 phosphorylation is dependent on mTORC1 activity, these results suggest an unknown mechanism for increasing 4EBP1 phosphorylation in hypoxia.

Polyamine incorporation by TG2 adds positive charges to the substrate proteins, and this modification leads to the changes in protein–protein interaction (Jeon et al, 2003a). Hypothesizing that the altered interaction between polyaminated 4EBP1 and mTORC1 may augment the phosphorylation of 4EBP1, we compared the ability of native and polyaminated 4EBP1 to bind to Raptor, a scaffolding protein in the mTORC1 that recruits substrates by binding (Wang et al, 2006). Pull-down with recombinant 4EBP1

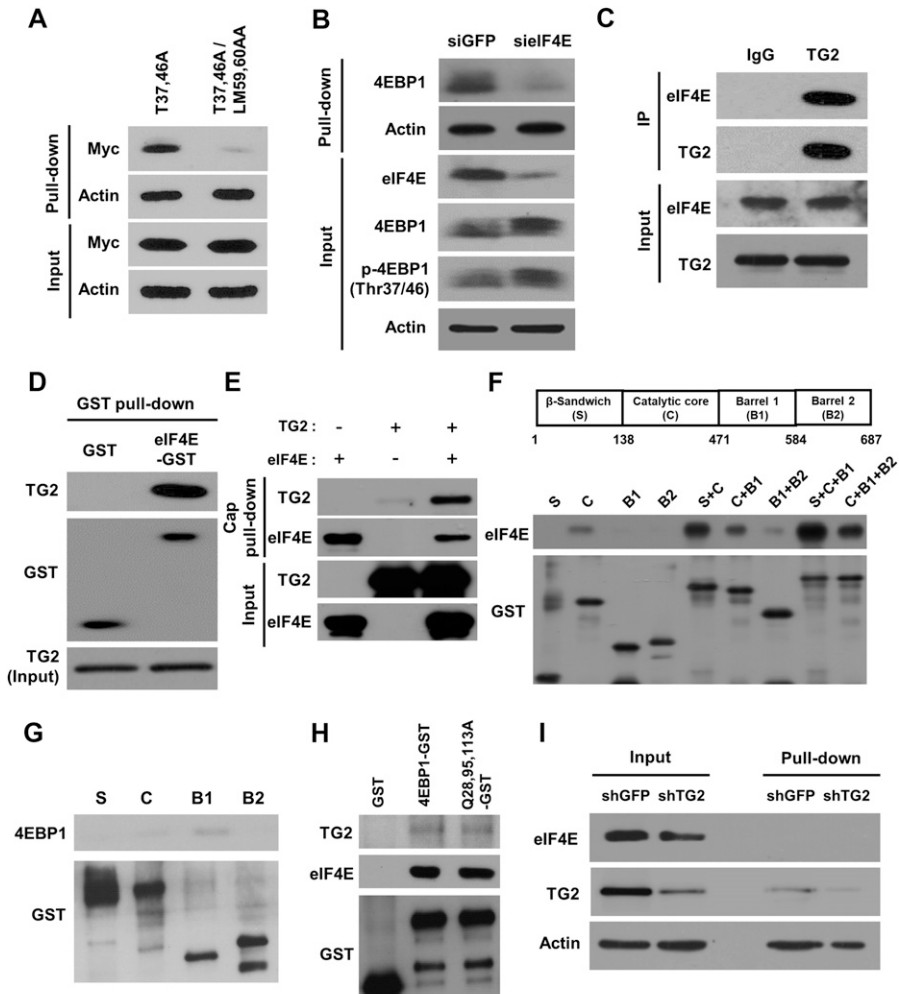

**Figure 3. TG2 preferentially polyaminates eIF4E-bound 4EBP1 in hypoxic cells.**
**(A)** Comparison of 4EBP1 polyamination between free and eIF4E-bound 4EBP1. A549 cells were transfected with phosphorylation-defective 4EBP1 (T37,46A) or eIF4E-binding–defective 4EBP1 (T37,46A/LM59,60AA) and then exposed to hypoxia (1% $O_2$) with BP (200 $\mu$M) for 6 h. After lysis, BP-incorporated 4EBP1 was detected by streptavidin pull-down and Western blot analysis. **(B)** Effect of eIF4E knockdown on 4EBP1 polyamination. A549 cells treated with sieIF4E RNA and exposed to hypoxia with BP for 6 h, and then cell lysates were pulled-down with streptavidin and then immunoblotted with anti-4EBP1 antibody. **(C)** Complex formation of TG2 and eIF4E. Lysates of A549 cells were immunoprecipitated with control and anti-TG2 antibodies and then immunoblotted with anti-TG2 or eIF4E antibody. **(D, E)** Direct association between TG2 and eIF4E. **(D, E)** Purified TG2 and eIF4E-GST were incubated for 1 h and association was estimated by pull-down using glutathione sepharose beads (D) or 7-methyl GTP sepharose beads (E). **(F)** Domain mapping of TG2 for eIF4E binding. Schematic diagram showing the domain organization of TG2 (upper panel). GST-tagged deletion mutants of TG2 were incubated with purified eIF4E, and eIF4E-bound domains were estimated by glutathione pull-down and Western blot analysis. **(G)** Domain mapping of TG2 for 4EBP1 binding. GST-tagged deletion mutants of TG2 were incubated with purified 4EBP1, and 4EBP1-bound domains were estimated by glutathione pull-down and Western blot analysis. **(H)** Binding affinity of polyamination-defective 4EBP1 for TG2 and eIF4E. Lysates of A549 cells were incubated with purified wild-type and mutant 4EBP1-GST, followed by streptavidin pull-down and immunoblot assays with anti-TG2 and eIF4E antibodies. **(I)** eIF4E was not polyaminated by TG2. A549 cells exposed to hypoxia (1% $O_2$) with BP (200 $\mu$M) for 6 h, and cell lysates were pulled-down with streptavidin and then immunoblotted with anti-TG2 or eIF4E antibody. TG2 was auto-modified by BP incorporation.

precipitated Raptor from hypoxic A549 and HeLa cell lysates, and polyamination by TG2 increased the level of Raptor (Figs 4D and E and S4). Moreover, TG2-mediated polyamination enhanced the phosphorylation of the recombinant 4EBP1 by mTORC1 from hypoxic cell lysate (Fig 4E). Consistent with the results that spermidine is the preferred substrate for 4EBP1 polyamination (Fig 1E), spermidinylated 4EBP1 effectively pulled-down Raptor and enhanced its phosphorylation by mTORC1, compared with putrescine or spermine-incorporated 4EBP1 (Figs 4E and S4). The increase in Raptor binding and phosphorylation of recombinant 4EBP1 by TG2-mediated polyamination were abrogated in polyamination-defective 4EBP1 (4EBP1$^{Q28,95,113A}$) (Fig 4F). Therefore, TG2-mediated polyamination of 4EBP1 increased the interaction between 4EBP1 and mTORC1, resulting in increased phosphorylation of 4EBP1 in hypoxia.

### TG2 enhances cap-dependent translation

Phosphorylation of 4EBP1 by mTORC1 is critical for inducing cap-dependent translation via the stimulation of eIF4F complex formation (Siddiqui & Sonenberg, 2015). We, thus, tested whether TG2 regulates cap-dependent translation using a bicistronic translation reporter that expresses chloramphenicol acetyltransferase (CAT) in

cap-dependent translation and luciferase in internal ribosome entry site (IRES)–dependent translation (Fig S5) (Yang et al, 2003). We introduced a bicistronic reporter into SK-N-SH cells, which generally show no detectable TG2 expression but exhibited an increase in intracellular TG2 activity after TG2 transfection (data not shown). The expression of TG2 in SK-N-SH cells showed a dose-dependent increase in CAT protein (Fig 5A). However, overexpression of TG2 had little effect on the expression of phosphatase and tensin homolog (PTEN) and activities of AKT and mTORC1 (Fig S6A). Under these conditions, treatment with cystamine, a TG inhibitor, decreased levels of CAT protein but increased relative luciferase activity (Fig 5B). Cystamine treatment on SK-N-SH cells with no TG2 expression slightly increased the luciferase activity, but little effect on cap-dependent translation was estimated by CAT expression (Fig S6B). Moreover, expression of active-site mutant TG2$^{C277S}$ failed to increase the level of CAT protein and had minimal effect on luciferase activity relative to wild-type TG2 (Fig 5C). Neither TG2 overexpression nor cystamine treatment had an effect on the level of luciferase mRNA measured by quantitative RT-PCR (data not shown), indicating that TG2 and cystamine did not impact the transcriptional processes. Together with the CAT results above, these experiments demonstrate that

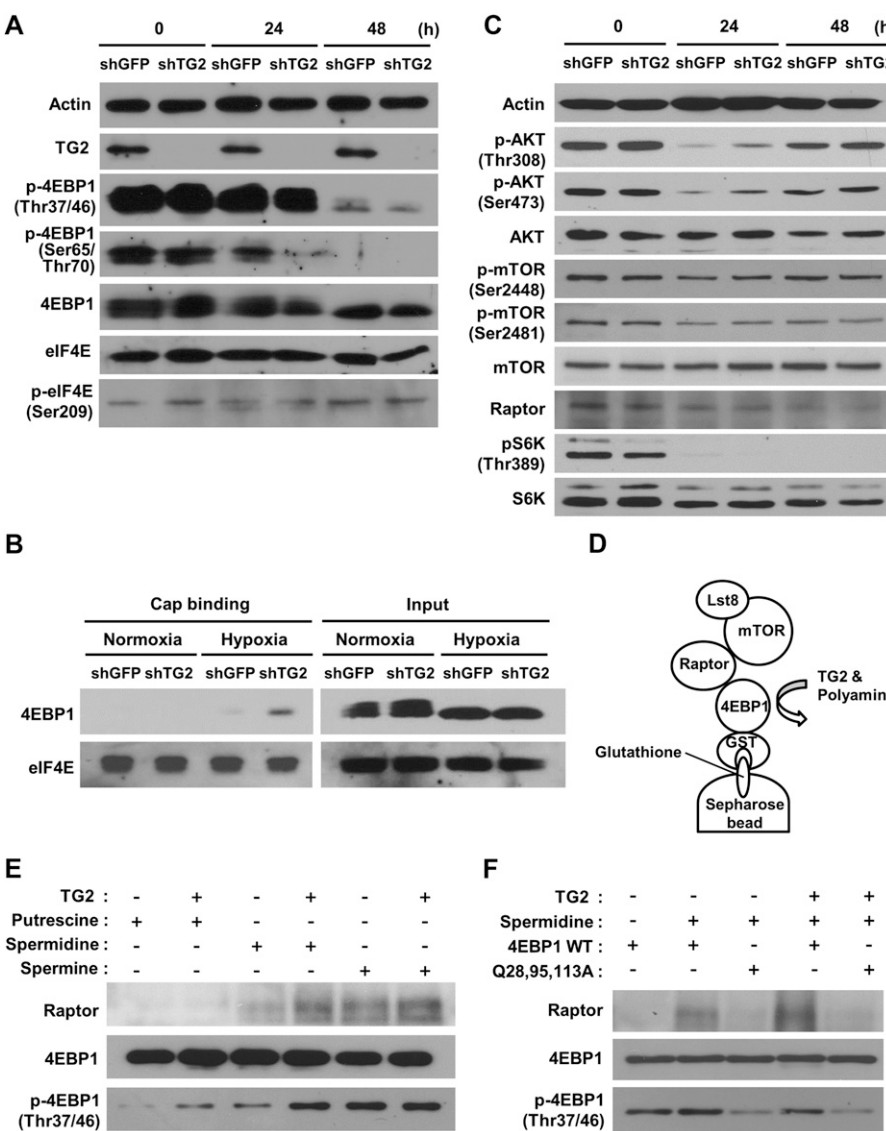

**Figure 4. 4EBP1 polyamination increases its phosphorylation by enhancing binding affinity for Raptor in hypoxia.**
**(A)** An increase of 4EBP1 phosphorylation by TG2 in hypoxic cells. Wild-type and TG2-knockdown A549 cells were exposed to hypoxia (1% $O_2$) for 48 h. Phosphorylation of 4EBP1 was evaluated by Western blot analysis. **(B)** A decrease of eIF4E-bound 4EBP1 by TG2 in hypoxia. A549 cells treated with shRNA for TG2 (shTG2) and exposed to hypoxia (1% $O_2$) for 24 h. Then cell lysates were pulled-down with 7-methyl GTP sepharose beads and immunoblotted with antibodies for 4EBP1 and eIF4E. **(C)** Changes in AKT and mTORC1 signaling in response to hypoxia. Wild-type and TG2-knockdown A549 cells were exposed to hypoxia (1% $O_2$) for 48 h. Phosphorylation of AKT, mTOR, and S6K were evaluated by Western blot analysis. **(D)** Schematic presentation of the assay for the binding affinity between polyaminated 4EBP1 and Raptor complex. **(E)** TG2-mediated 4EBP1 polyamination increases its binding affinity for Raptor and its phosphorylation by mTORC1 in vitro. 4EBP1-GST attached to glutathione sepharose beads was incubated with TG2 and polyamines (500 $\mu$M for 1 h) and then with A549 cell lysates which were exposed to hypoxia (1% $O_2$, for 6 h) before incubation. Raptor and phosphorylated 4EBP1 (p-4EBP1) were detected by Western blot analysis. **(F)** Polyamination-defective 4EBP1 abrogates the increased Raptor binding and phosphorylation by TG2-mediated polyamination of 4EBP1. Wild-type (WT) and polyamination-defective (Q28,95,113A) 4EBP1-GST attached to glutathione sepharose beads was incubated with TG2 and spermidine (500 $\mu$M for 1 h) and then with A549 cell lysates which were exposed to hypoxia (1% $O_2$, for 6 h) before incubation. Raptor and phosphorylated 4EBP1 (p-4EBP1) were detected by Western blot analysis.

TG2 increases cap-dependent translation, but not transcriptional processes, in an activity-dependent manner.

We next examined the effect of 4EBP1 expression on cap-dependent translation. As expected from earlier results, overexpression of TG2 increased CAT when the cells were co-transfected with wild-type 4EBP1 (Fig 5D), whereas transfection of polyamination-defective 4EBP1[Q28,95,113A] exhibited decreased CAT that was not ameliorated by co-transfection with TG2 (Fig 5E), confirming the role of TG2-mediated 4EBP polyamination in enhancing cap-dependent translation. Oxidative stress, treatment with chemotherapeutics, hypoxia, and UV irradiation are known activators of TG2 via increased intracellular calcium (Jeong et al, 2009; Jang et al, 2010; Cho et al, 2012); therefore, we assessed the effect of a calcium ionophore on cap-dependent translation. In MCF7 cells treated with A23187, similar results were obtained with TG2 and 4EBP1 overexpression and cystamine treatment (Fig S7A–C). However, TG2-overexpressing HEK293 cells did not differ significantly in the amount of newly synthesized proteins, as assessed by S[35]-methionine incorporation, although TG2-overexpressing

cells did show slightly elevated levels after 18 h of A23187 treatment (Fig 5F). Thus, TG2 seems to induce qualitative changes in the proteome through 4EBP polyamination.

### TG2 mediates selective mRNA translation in hypoxia

It has been suggested that 4EBPs limit eIF4E availability under mTORC1-inhibited conditions, leading to a bottleneck for eIF4E availability, and thereby resulting in selective mRNA translation (De Benedetti & Graff, 2004). To investigate whether polyamination of 4EBPs may mediate this selective translation, we identified the proteins affected by TG2 activity in a translation-dependent manner. To this end, we metabolically labeled newly synthesized proteins using azidohomoalanine (AHA), an amino acid analog of methionine that can be purified by a covalent conjugation of azide–alkyne cycloaddition click chemistry reaction (Dieterich et al, 2006). We established a quantitative proteomics approach using

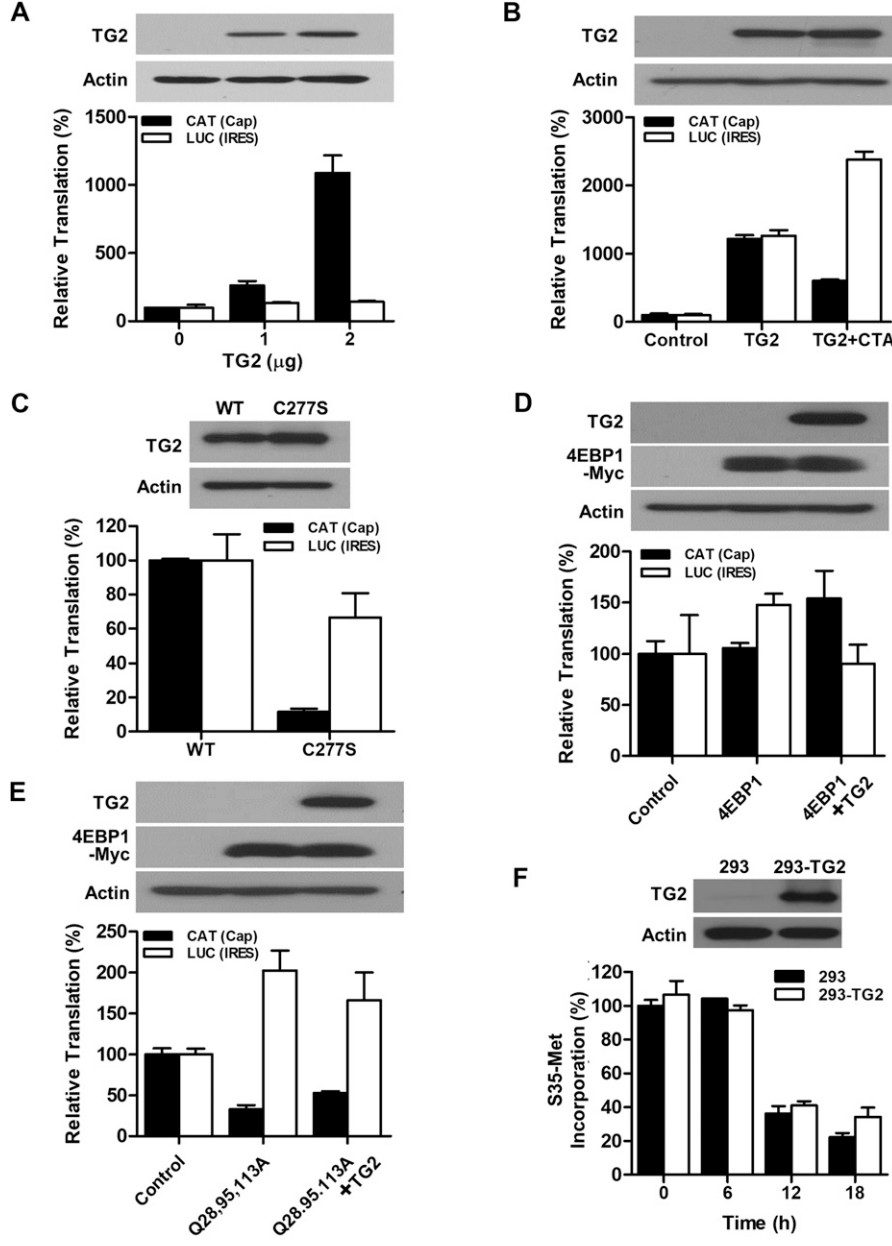

**Figure 5. TG2 enhances cap–dependent translation.**
**(A, B, C, D, E)** In normoxic condition, SK-N-SH cells were transiently transfected with pSG5-TG2 and pcDNA-CAT/EMCV/LUC (0.2 $\mu$g), which reports cap- and IRES-dependent translation via expression of chloramphenicol acetyltransferase (CAT) and luciferase (LUC), respectively. **(A, B, C, D, E)** Effect of TG2 overexpression (A), treatment with cystamine, a TG2 inhibitor (500 $\mu$M, CTA; B), expression of active site–mutant TG2 (C227S; C), 4EBP1 expression (D), and expression of polyamination-defective 4EBP1 (Q28,95,113A; E) on reporter activities was assessed by measuring CAT with ELISA and luciferase activity assay after normalizing protein amount. **(B, C, D, E)** Two micrograms of TG2 vector were applied for the experiments in (B, C, D, E). The expression of wild-type and mutant TG2 and 4EBP1 were evaluated by Western blot analyses. **(F)** Effect of TG2 overexpression on total protein synthesis. The control and stably TG2-overexpressed HEK293 cells were treated with 5 $\mu$M A23187 for 0–18 h and pulse-labeled for 20 min in the methionine-free medium with 10 $\mu$Ci [$^{35}$S]-methionine. After protein precipitation, radioactivity was measured by liquid scintillation counting. The incorporation of [$^{35}$S]-methionine was depicted compared with control HEK293 cells at 0 h. The data represent the mean values ± standard deviations based on experiments that were performed in triplicate.

label-free quantification for newly synthesized AHA-containing proteins purified with streptavidin beads.

Cells in AHA-containing media were exposed to hypoxia in the absence (Hyp) and presence of cystamine (Hyp+Cys). A total of 2,072 and 1,826 proteins were identified with at least one unique and quantified peptide in Hyp and Hyp+Cys conditions, respectively. Of these, 375 proteins were found only in Hyp (342 proteins) or significantly increased in Hyp (33 proteins; fold-change > 2, $P$ < 0.05), a total of 18.1% of Hyp proteins (Fig 6A and B and Table S1). One hundred eighty-nine proteins were found only in Hyp+Cys (96 proteins) or significantly increased in Hyp+Cys (93 proteins; fold-change > 2, $P$ < 0.05), a total of 10.4% of Hyp+Cys proteins (Fig 6A and B and Table S2).

Gene ontology analysis of these proteins in the DAVID bio-informatics resources (http://david.ncifcrf.gov) (Jiao et al, 2012)

revealed that under hypoxic conditions, TG2 promotes the translation of genes involved in cellular adhesion, macroautophagy regulation, and endoplasmic reticulum organization (Fig 6C and Table S3) but suppresses the translation of genes involved in termination of RNA polymerase II transcription, mRNA 3′-end processing, and multivesicular body assembly (Fig 6D and Table S4). Hallmark gene set analysis from Molecular Signature Database (MSigDB; http://software.broadinstitute.org/gsea/msigdb/index.jsp) (Liberzon et al, 2015) showed that gene sets such as "HALLMARK_UNFOLDED_-PROTEIN_RESPONSE," "HALLMARK_GLYCOLYSIS," "HALLMARK_A-PICAL_JUNCTION," and "HALLMARK_DNA_REPAIR" were enriched only in the Hyp condition ($P$ < 0.05; Tables S5 and S6), and gene sets such as "HALLMARK_ALLOGRAFT_REJECTION" and "HALL-MARK_KRAS_SIGNALING_UP" were enriched only in the Hyp+Cys

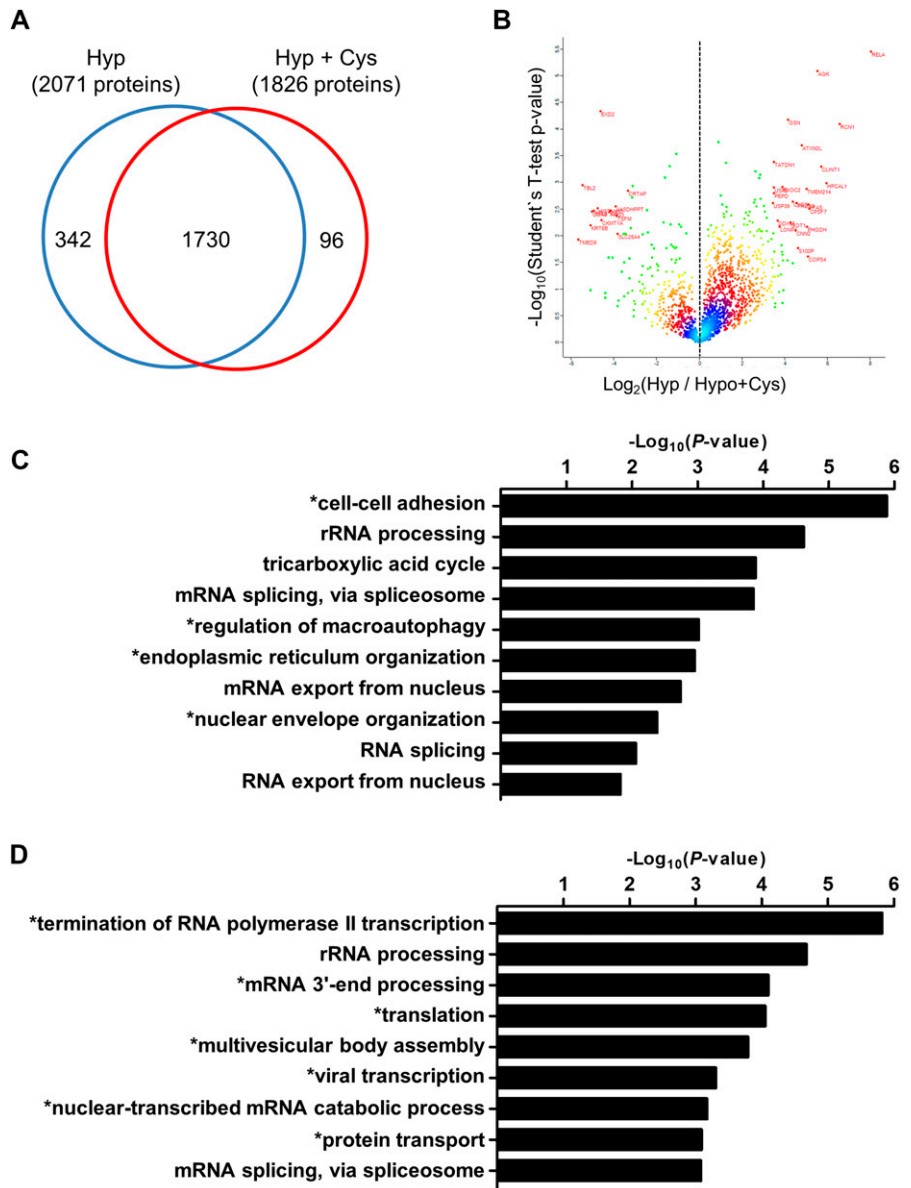

**Figure 6. TG2 mediates selective mRNA translation in hypoxia.**
**(A)** Venn diagram of the numbers of proteins identified and quantified from A549 cells cultured under hypoxia (Hyp) and hypoxia with cystamine treatment (500 $\mu$M; Hyp + Cys). **(B)** Volcano plot of protein expression profiles showing proteins that were up-regulated or down-regulated by TG2 inhibition (by cystamine treatment) under hypoxic conditions, with the $\log_2$ value of ratio (Hyp/Hyp + Cys) on the x-axis and $\log_{10}$ value of $P$-values on the y-axis. Differences were assessed by two-sided $t$ test. **(C, D)** Gene ontology (GO) analysis of proteins qualitatively regulated by TG2 activity in hypoxia. **(C, D)** Enriched GO terms in Hyp (C) and Hyp + Cys condition (D) are represented with $P$-values. $P$-values were estimated by the DAVID bioinformatics resources (http://david.ncifcrf.gov). **(C, D)** Asterisks indicate GO terms exclusively detected only in Hyp (C) or Hyp + Cys condition (D).

condition ($P < 0.05$; Tables S5 and S6). These results indicate that TG2 differentially regulates the translation of selected sets of transcripts that together comprise more than 26% of newly synthesized proteins under hypoxic conditions.

### TG2 enhances translation of mRNAs with GC-rich 5′UTR under hypoxia

The 5′UTRs of mRNAs play a role in the selective translation of mTOR-inhibited cells (Leppek et al, 2018). To test whether 5′UTR features are implicated in TG2-mediated translational control, we compared 5′UTR sequences and structures between mRNAs corresponding with the proteins uniquely identified under Hyp (342 proteins) and Hyp+Cys conditions (96 proteins). We found no differences in 5′UTR length between the two groups, but %GC and ΔG in the 5′UTR of Hyp proteins was significantly higher and lower, respectively, than those of Hyp+Cys proteins (Fig 7A). In addition, the presence of an upstream ORF (uORF) motif or 5′ terminal oligo-pyrimidine (TOP) sequence was not associated with TG2-mediated selective translation (Fig 7B). These results suggest that TG2 may be also involved in resolving the GC-rich structure of 5′UTR in conjunction with 4EBP polyamination, thereby contributing to selective increase in translational efficiency. Moreover, among the gene ontology affected by TG2, macroautophagy-related proteins were down-regulated by cystamine treatment under hypoxic conditions (Table S3). We examined the effect of TG2 inhibition on these protein levels under normoxic conditions. Unexpectedly, cystamine treatment enhanced protein levels of SNX6, UBQLN2, CASP3, and

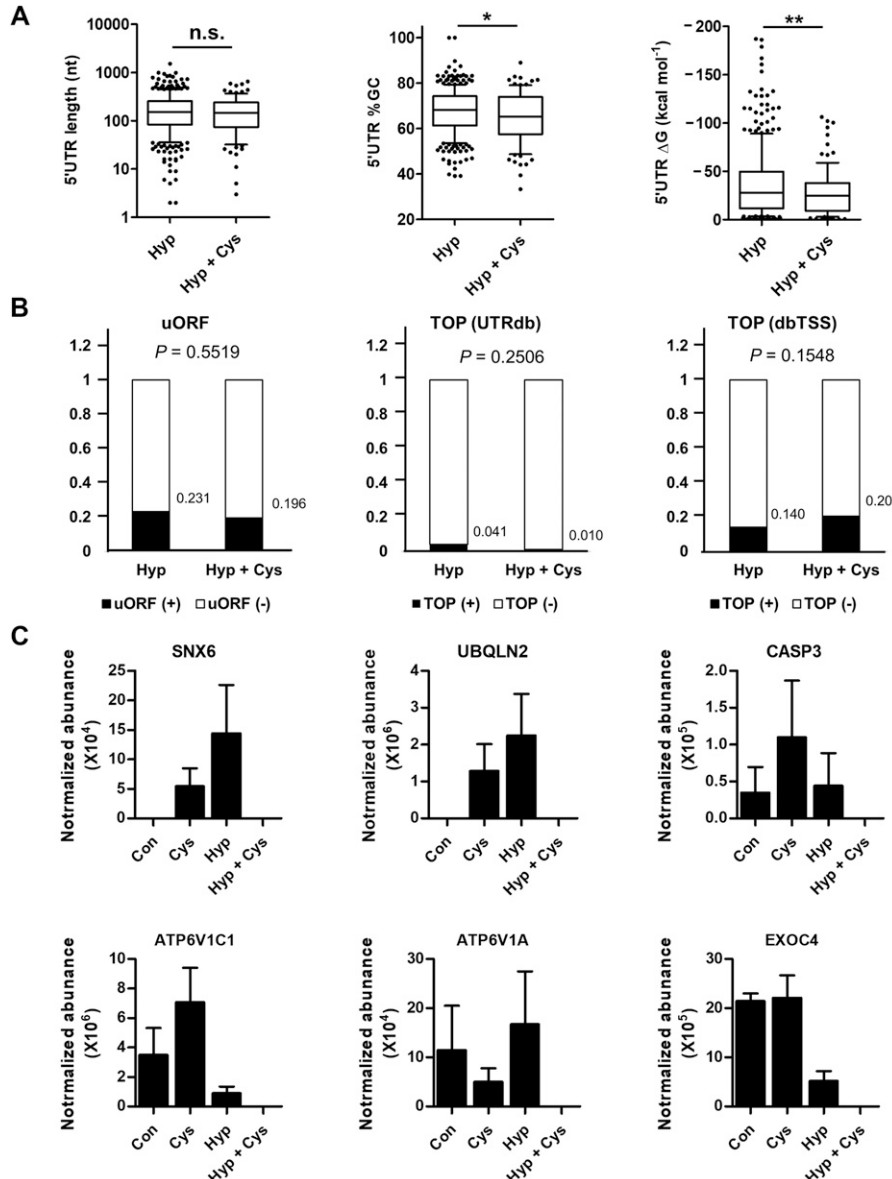

**Figure 7. TG2 enhances translation of mRNA with GC-rich 5′UTR in hypoxia.**
**(A, B)** Length of 5′UTR, %GC, and ΔG (A) and the presence of upstream ORF and terminal oligopyrimidine (from UTRdb and dbTSS; B) were compared between mRNAs for corresponding to proteins identified under hypoxia (Hyp) and hypoxia with cystamine treatment (Hyp + Cys) conditions. **(A, B)** Differences were assessed by two-sided $t$ test (A) and chi-square test (B). Asterisks indicate significant differences (*$P < 0.05$; **$P < 0.01$). **(C)** Normalized abundance of representative macroautophagy-related proteins enriched in hypoxia. Effect of cystamine treatment on macroautophagy-related protein levels was compared between cells cultured under normoxic and hypoxic conditions. The data represent the mean values ± standard deviations based on experiments that were performed in triplicate. Cys, cystamine; Hyp, hypoxia.

ATP6V1C1, but it suppressed ATP6V1A and showed little effect on EXOC4 (Fig 7C), indicating that TG2-mediated translational control is hypoxia specific.

# Discussion

Cancer cells reprogram their gene expression in response to hypoxic stress mainly through HIF1-mediated transcriptional regulation to adapt to adverse microenvironments. This reprogramming leads to changes in a variety of functions, including glucose and energy metabolism, proliferation, apoptosis, invasion, and metastasis (Balamurugan, 2016). Despite the suppression of overall mRNA translation due to ATP depletion and subsequent mTORC1 inhibition, translation of mRNAs essential for hypoxic responses is paradoxically enhanced (Spriggs et al, 2010), suggesting that the signaling pathway activated by hypoxic stress controls translational processes. In this study, we investigated the role of TG2, a HIF1-dependent enzyme, in the regulation of mRNA translation and report a new pathway that links hypoxia to cap-dependent translation. TG2 incorporates polyamines into 4EBPs in response to hypoxic stress, and 4EBP1 polyamination enhances binding affinity of 4EBP1 for mTORC1, thereby increasing phosphorylation of 4EBP1 and cap-dependent translation. Moreover, TG2 activity augments protein synthesis of a set of mRNAs that adjust cells to hypoxic stress. Our results indicate that TG2 is a critical translational regulator in the adaptation to hypoxia.

Protein synthetic activity is mainly regulated by mTORC1, which integrates cellular microenvironmental signals such as growth factors, nutrients, and oxygen availability (Siddiqui & Sonenberg, 2015). Under hypoxic conditions, mTORC1 is suppressed by the activation of the AMPK signaling pathway, but the translation of selected mRNA is

enhanced to support hypoxic cell survival. Mechanistically, mTORC1 activity increases eIF4E availability by phosphorylation of 4EBP, enhancing cap-dependent translation. Phosphorylation of 4EBP depends on its binding to Raptor, a scaffolding protein of mTORC1, through a conserved set of five C-terminal amino acids (FEMDI) known as the TOR signaling motif (Zoncu et al, 2011). Although all TOR signaling residues are known to be necessary for Raptor binding, a recent report reveals that the preceding Gln113, along with Phe114, participates in binding to a pocket by making a $\pi$–$\pi$ interaction between the side chains of the two residues. Mutation of Gln113 to Ala results in a decrease of its affinity for Raptor by 95% compared with wild-type 4EBP1 (Yang et al, 2017). Indeed, our 4EBP mutant analyses identified Gln113 as a major polyamination site and showed that 4EBP1 polyamination increased its binding affinity for Raptor, suggesting that polyamination of Gln113 residue reinforces the interaction with Phe114 or with the binding groove of Raptor. Moreover, our results showed that TG2 preferentially polyaminates eIF4E-bound 4EBP1. Together, these results provide a novel mechanism for increasing eIF4E availability under mTORC1-suppressed conditions.

Polyamines are positively charged small molecules and the concentration of intracellular polyamines was near 1 mM (Igarashi & Kashiwagi, 2010). Although most of them were bound to nucleic acid, protein, and phospholipid, the concentration of free polyamines was about 20–200 μM. Although the Km values for TG2-mediated polyamination of 4EBPs were high, we suggest that 4EBPs can be modified by TG2 in cellular context because of the high concentration of intracellular free polyamines. Our data also showed that only spermidine treatment without TG2 increased the interaction between 4EBP1 and Raptor (Fig 4E and F). This result was probably due to the enhanced electrostatic protein–protein interactions by positively charged polyamines (Thomas et al, 1999; Berwanger et al, 2010). The polyamine-mediated modulation of the interaction between 4EBP1 and Raptor needs to be further evaluated.

In our experimental condition, hypoxia decreased the expression levels of AKT and S6K, especially after 48 h (Fig 4C). Previous reports showed that AKT and S6K were substrates of caspase 3 (Rokudai et al, 2000; Cho et al, 2017), and hypoxia is one of stresses causing caspase 3 activation (Malhotra et al, 2001). Therefore, AKT and S6K are possibly cleaved by caspase activation in hypoxic conditions. However, exact molecular mechanisms of the regulation of AKT and S6K expression in hypoxia need to be further studied.

Various mTORC1-independent translational mechanisms have previously been reported to explain protein synthesis in hypoxic cells, including IRES-mediated translation (Svitkin et al, 2005), and HIF2$\alpha$-dependent translation by binding of the HIF2$\alpha$–RBM4–eIF4E2 complex to RNA hypoxia-response element (Uniacke et al, 2012). Both mechanisms depend on cis-acting elements located at the 5′UTR or 3′UTR of RNA and account for proteins synthesized under TG2-inhibited conditions in hypoxic cells (Fig 6A and B). By contrast, the TG2-mediated mechanism described here promotes the efficiency of canonical translational machinery. Moreover, our proteomic analysis revealed that the TG2-induced proteome exhibits a higher complexity and quantity relative to the TG2-independent proteome. Thus, these results suggest that the HIF1–TG2–4EBP axis may be a major pathway responsible for enhancing cap-dependent translation in hypoxic cells.

4EBP1 suppresses cellular proliferation by inhibiting eIF4E-mediated protein synthesis, suggesting that 4EBPs function as tumor suppressors.

By contrast, it has been reported that 4EBP1 is also involved in promoting tumor progression by enhancing the translation of a selected subset of mRNA, such as c-Myc, Cyclin D1, HIF1, and VEGF, especially under stress conditions (Musa et al, 2016). Indeed, 4EBP1-knockdown in glioblastoma cells show increased cell death in response to hypoxic stress (Dubois et al, 2009). The mechanism for this paradoxical 4EBP1 activity remains elusive. In this study, our proteomic analysis of newly synthesized proteins in hypoxic cells revealed that TG2 inhibition selectively down-regulated proteins associated with cell adhesion, macroautophagy, and RNA metabolism (Fig 6C), implicating TG2 in the selective mRNA translation. TG2-mediated polyamination of 4EBPs increased eIF4E availability in hypoxic cells, which, in turn, enhances cap-dependent translation. Moreover, this pathway also explains how the translation of a subset of mRNAs is preferentially up-regulated. Translation efficiency is known to be affected by the 5′UTR structures of mRNA, such as uORF or 5′TOP sequences, which, respectively, attenuates translation by inducing leaky scanning (Hinnebusch et al, 2016) or selectively enhances translation efficiency when eIF4E availability is increased (Thoreen et al, 2012). Our results showed that TG2-mediated selective mRNA translation is not associated with 5′UTR length or the presence of uORF or 5′TOP, but with high GC content, which requires RNA helicase activity for scanning the structured 5′UTR (Hinnebusch et al, 2016). Because free eIF4E makes complex with RNA helicase eIF4A for enhancing translation initiation (Siddiqui & Sonenberg, 2015), these data support the role of TG2 in enhancing translational efficiency of selected mRNA.

TG2 regulates the activity of substrate proteins in response to various stressors through polyamination, crosslinking, or deamidation modification, conferring a growth advantage on cancer cells (Han & Park, 1999; Mehta et al, 2004; Fok et al, 2006; Herman et al, 2006; Verma et al, 2006). In metastatic tumors found in lymph node and drug-resistant cancers of several origins, an increased TG2 level is associated with acquisition of epithelial–mesenchymal transition and stem cell–like characteristics, traits that require changes in gene expression (Kumar et al, 2010, 2011). Previous reports showed that TG2 regulates gene transcription by modulating the activity of transcription factors such as NF-$\kappa$B (Mann et al, 2006; Jang et al, 2010). TG2 can also control protein quantity at the posttranslational level by modifying and sequestering proteins such as caspase 3, BAX-binding protein, and nucleophosmin (Yamaguchi & Wang, 2006; Park et al, 2008; Jang et al, 2010). Moreover, our results in this study demonstrated that TG2 is involved in the translational control of mRNAs in cells under hypoxic conditions. Therefore, TG2 is a critical mediator in the regulation of gene expression at multiple levels that contributes to the hypoxia-induced reprogramming.

In summary, we demonstrated that TG2 plays a critical role in reprogramming gene expression in response to hypoxic stress by increasing mTORC1-mediated phosphorylation of 4EBP1 and cap-dependent translation. Given the prevalence of hypoxia in tumorigenesis and the utility of these adaptations for tumors to overcome cellular crowding, TG2 is a promising target for the effective treatment of cancers.

# Materials and Methods

### Cell culture and generation of TG2 knockdowned cell line

A549 and SK-N-SH cells were maintained in RPMI 1640 medium (WelGENE) and Dulbecco's modified Eagle medium (WelGENE)

containing 10% heat-inactivated fetal bovine serum, respectively. Penicillin (100 U/ml; Invitrogen), and streptomycin sulfate (100 μg/ml; Invitrogen) were added to all culture media. All cells were maintained in a humidified incubator with 5% $CO_2$ at 37°C and incubated under either normoxic (20% $O_2$) or hypoxic (1% $O_2$ balanced with nitrogen) conditions for indicated period of time.

A549 cell line that down-regulate TG2 (A549[shTG2]) was established as described previously (Cho et al, 2010). In brief, the cells were transfected with pSUPER plasmid containing shRNA for TG2 or GFP (A549[shGFP]) and selected with 800 μg/ml of geneticin (Invitrogen) for 2 wk. The target sequences for TG2 and GFP were 5′-GGGCGAACCACCTGAACAA-3′ and 5′-GCAAGCTGACCCTGAAGTTC-3′, respectively (Kim et al, 2005; Cho et al, 2010). The knockdown of TG2 was confirmed by Western blot analysis.

### Analysis of TG2 substrates in hypoxic cells

TG2 substrate proteins in hypoxic condition were identified as described previously (Jang et al, 2010). A549 cells were labeled with 0.2 mM 5-BP (Pierce) under hypoxic condition (1% $O_2$) for 48 h. The harvested cells were resuspended in lysis buffer (50 mM Tris–Cl, pH 7.5, 150 mM NaCl, 1% Triton X-100, 0.02% SDS, 0.5% sodium deoxycholate, 1 mM EDTA, and protease inhibitor cocktail [Roche]). After dialysis against PBS overnight at 4°C, the cell lysates were attached to streptavidin-conjugated magnetic beads (Dynal Biotech) on a rocking platform for 3 h at 4°C. The beads were washed with PBS-T (PBS containing 1% Tween-20). The proteins bound to the beads were eluted by boiling for 10 min in sample loading buffer (12 mM Tris–Cl, pH 6.8, 5% glycerol, 0.4% SDS, 1% 2-mercaptoethanol, and 0.02% bromophenol blue) and subjected to SDS–PAGE for Western blot analysis.

To investigate the effect of phosphorylation status of 4EBP1 on TG2-mediated modification, cell lysate in buffer (50 mM Tris–Cl, pH 7.5, 150 mM NaCl, 0.5% Triton X-100, and protease inhibitor cocktail) was treated with calf intestinal phosphatase (New England Biolabs) for 4 h at 37°C and incubated with 1 mM BP in the presence of 5 mM $CaCl_2$ and 1 mM DTT for 1 h at 37°C. After TG reaction was stopped by 5 mM EDTA, BP-incorporated proteins were purified with streptavidin-conjugated magnetic beads and analyzed with Western blot analysis, as above.

### Purification of GST fusion proteins and GST pull-down assay

Recombinant 4EBP1-GST fusion protein and GST protein were expressed in BL21 cells by 1 mM IPTG (Amnesco). Bacteria were sonicated and centrifuged at 25,000*g*, and the supernatant was incubated with glutathione sepharose resins (GE Healthcare) for 1 h at 4°C. The attached proteins were eluted by elution buffer (50 mM Tris–Cl, pH 8.0, and 10 mM reduced glutathione [Sigma-Aldrich]). The purity of recombinant proteins was checked by Coomassie stain and Western blot analysis with anti-GST antibody. For GST pull-down, GST fusion proteins attached to glutathione sepharose were resuspended in polysome buffer (50 mM Tris–Cl, pH 7.5, 50 mM KCl, 5 mM $MgCl_2$, and 1 mM DTT), and cell lysates or purified proteins were incubated with attached proteins for 1 h by rotating. Proteins were eluted in sample loading buffer by boiling for 10 min and detected by Western blot analysis.

### Detection of TG2-modified substrates

Modification of 4EBP1 by TG2 was verified by detecting incorporated BP as a probe. For solid-phase microtiter plate assay (Slaughter et al, 1992), the purified GST and 4EBP1-GST fusion protein were diluted to coating buffer (50 mM Tris–Cl, pH 7.5, 150 mM NaCl, 5 mM EGTA, and 5 mM EDTA). The samples were added to each well of a 96-well microtiter plate (Nunc) and incubated for 2 h at 37°C. After blocking with 3% BSA in PBS-T, 1 mM BP and TG2 in TG reaction buffer (50 mM Tris–Cl, pH 7.5, 10 mM $CaCl_2$, 0.5% Triton X-100, and 1 mM DTT in 2× TBS) were added and further incubated for 1 h at 37°C for TG reaction. After washing with PBS-T, streptavidin coupled to horseradish peroxidase (1:1,000; Jackson Laboratory) in PBS-T with 3% BSA was added to each well and incubated at room temperature for additional 1 h. For colorimetric measurement, substrate solution (0.4 mg of o-phenylenediamine dihydrochloride/ml of 50 mM sodium citrate phosphate, pH 5.0) was added to each well, and, after 15 min, the reactions were stopped by adding 1M $H_2SO_4$. Incorporation of BP was quantitated by measuring the absorbance at 490 nm on microplate spectrophotometer (Molecular Devices). The $K_m$ values of TG2 enzyme reaction for BP were also calculated using Prism 4.0. To remove BP incorporation to GST, all glutamine residues (Gln-15, -67, -188, -204, and -207) present in GST were substituted with asparagines as previously described (Sugimura et al, 2006). Incorporation of BP to 4EBP1 was also detected by Western blot analysis using strepavidine coupled to horseradish peroxidase and chemiluminescent reaction. For polyamine competition assay, TG2-mediated BP incorporation was performed in the presence of putrescine, spermine, or spermidine.

Cross-linking of TG2 substrates was tested using Coomassie stain, and casein was used as positive control. Purified proteins (GST, 4EBP1-GST, and casein) were incubated with purified TG2 for 1 h in TG reaction buffer and boiled in sample loading buffer for 10 min. After separating on SDS–PAGE, the samples on gel were detected by Coomassie blue stain.

To detect in situ modification of substrates, the cells were incubated for 1 h with 1.5 mM BP and harvested by centrifugation. The cell lysates were prepared in lysis buffer, followed by centrifugation (14,000*g*, 10 min at 4°C). After quantitating the protein concentration, the BP-incorporated proteins were purified using streptavidin coupled to sepharose resins and detected by Western blot analysis.

### In situ TG activity assay

In situ TG activity was assayed by estimating the amount of 5-BP that is incorporated into the cellular proteins. The cells were incubated for 1 h with 1.5 mM BP and harvested by centrifugation. The cell lysates were prepared in lysis buffer, followed by centrifugation (14,000*g*, 10 min at 4°C). After quantitating the protein concentration, the protein samples were boiled in sample loading buffer for 10 min, separated on SDS–PAGE, and transferred onto nitrocellulose membranes. The BP-incorporated proteins were probed using streptavidin coupled to horseradish peroxidase and detected by chemiluminescent reaction and exposure to X-ray film.

### Western blot analysis

Cells were lysed in buffer (50 mM Tris–Cl, pH 8.0, 150 mM NaCl, 0.1% SDS, 1% Triton X-100, 0.5% sodium deoxycholate, protease inhibitor cocktail, and phosphatase inhibitor cocktail [Roche]) and centrifuged at 20,000*g* for 30 min at 4°C. After quantitating the protein concentration in each cell extract, all protein samples were boiled

in sample loading buffer for 10 min, separated on SDS–PAGE, and transferred electrophoretically onto nitrocellulose membranes. The membranes were washed with TBS-T (TBS containing 0.1% Tween-20) for 5 min and blocked for 1 h with 5% skim milk in TBS-T. Then the membranes were probed for 2 h with anti-TG2 (Jeon et al, 2003b), anti-4EBP1 (Cell Signaling), anti-p-4EBP1 (Thr 37/46; Cell Signaling), anti-p-4EBP1 (Ser65/Thr70; Santa Cruz Biotechnology), anti-eIF4E (Santa Cruz Biotechnology), anti-p-eIF4E (Ser209; Cell Signaling), anti-AKT (Cell Signaling), anti-p-AKT (Ser473; Cell Signaling), anti-mTOR (Cell Signaling), anti-p-mTOR (Ser2448; Cell Signaling), anti-p-mTOR (Ser2481; Cell Signaling), anti-Raptor (Cell Signaling), anti-S6K (Cell Signaling), anti-p-S6K (Thr389; Cell Signaling), anti-GST (Santa Cruz Biotechnology), anti-Myc (Cell Signaling), and anti-Actin (Sigma-Aldrich) antibodies. After washing three times with TBS-T, the membranes were incubated with a 1:2,000 diluted secondary antibody coupled to horseradish peroxidase (Pierce) for 1 h, and washed five times with TBS-T. The immunoreactive proteins were visualized by incubation with SuperSignal chemiluminescent reagent (Pierce) for 5 min and exposure to X-ray film.

### Site-directed mutagenesis

Site-directed mutagenesis of 4EBPs was performed according to QuikChange site-directed mutagenesis kit (Stratagene) instruction manual. The primers for 4EBP1 mutation were as follows: 5′-GCA-GCTGCAGCGCGACCCCAAGCCG-3′ for Q9A, 5′-GCGACGGCGTGGCGCTC-CCGCCCGG-3′ for Q28A, 5′-CATGGAAGCCAGCGCGAGCCACCTGC-3′ for Q95A, 5′-CGGTGAAGAGTCAGCGTTTGAGATGGAC-3′ for Q113A, 5′-GACTA-CAGCACGGCCCCCGGCGGCACG-3′ for T37A, 5′-CTCTTCAGCACCGCCCCGG-GAGGTACC-3′ for T46A, and 5′-CATCTATGACCGGAAATTCGCGGCGGAGTG-TCGGAACTCACC-3′ for LM59,60AA. The primers for 4EBP2 mutation were as follows: 5′-CATCAGCGACGCCGCGGCACTACCTCATGACTATTG-3′ for Q29A and 5′-GTTGGGGATGATGCTGCGTTCGAGATGGACATC-3′ for Q115A. The primers for 4EBP3 mutation were as follows: 5′-GGGGCCGGGACGCACTGCCCGACTG-3′ for Q15A and 5′-GAGATACCCGATGACGCAGCCTTTGAAATGGACATCTAAC-3′ for Q95A. The mutated sequences are underlined.

### Co-immunoprecipitation

Cells were washed twice with ice-cold PBS and lysed in lysis buffer (50 mM Tris–Cl, pH 8.0, 150 mM NaCl, 1% Triton X-100, and protease inhibitor cocktail [Roche]) for 30 min. The cell lysates were centrifuged at 12,000$g$ for 10 min at 4°C, and supernatants were incubated with anti-TG2 or normal IgG antibody at 4°C for 4 h. Then, the samples were incubated with protein G–conjugated magnetic beads (Pierce) at room temperature for 10 min. The beads were washed four times with lysis buffer. Proteins were eluted in sample loading buffer by boiling for 10 min and detected by Western blot analysis.

### 7-methyl GTP affinity chromatography

To purify eIF4E and 4EBP1 protein complex, cells were lysed in cap lysis buffer (20 mM Tris–Cl, pH 7.4, 150 mM NaCl, 5 mM EDTA, 1% Triton X-100, protease inhibitor cocktail [Roche], and phosphatase inhibitor cocktail [Roche]). After centrifugation, the supernatant was incubated with 7-methyl GTP sepharose resin (GE Healthcare) at 4°C for 1 h and washed twice with cap lysis buffer. Proteins were eluted in sample loading buffer by boiling for 10 min and detected by Western blot analysis.

### Binding of mTOR complex 1 (mTORC1) to 4EBP1 affinity resins

Binding affinity of TG2-modified 4EBP1 to mTORC1 was estimated as previously described with modification (Wang et al, 2006). Recombinant 4EBP1-GST fusion protein was purified using glutathione sepharose resins, as above. 4EBP1-GST attached to resins was resuspended in TG reaction buffer and incubated with or without TG2 and 1 mM polyamines (putrescine, spermine and spermidine) for 1 h at 37°C. TG reaction was stopped by adding 10 mM EDTA. After washing with complex lysis buffer (10 mM sodium phosphate, pH 7.4, 1 mM EGTA, 1 mM EDTA, 1 mM DTT, and 0.1% Tween-20), cell extract in complex lysis buffer was added to resin and incubated on a rocking platform for 12 h at 4°C. The resins were washed four times with complex lysis buffer, twice with complex lysis buffer plus 0.5 mM NaCl, and then once with washing buffer (50 mM Tris–HCl, pH 7.4, 1 mM EDTA, and 1 mM EDTA). The relative amounts of Raptor attached to the resins were determined by Western blot analysis.

### Bicistronic translation reporter assay

In vivo translation assay of bicistronic reporter system was performed as previously described using pcDNA-CAT/EMCV/LUC plasmid (Fig S5), from which CAT and firefly luciferase were expressed by cap- and IRES-dependent mechanisms, respectively (Yang et al, 2003). The pcDNA-CAT/EMCV/LUC plasmid was transfected with or without TG2- and 4EBP1-expressing vectors, and the cells were harvested after 24 h for CAT and luciferase assay. CAT protein was quantitated with CAT ELISA (Roche) and firefly luciferase activity was measured by Luciferase assay system (Promega) according to instruction manual. All data were normalized to protein amount.

### Estimation of [$^{35}$S]-methionine incorporation

Protein synthesis was estimated by measuring incorporation of [$^{35}$S]-methionine to cellular proteins. Cells were treated with 5 $\mu$M A23187 for indicated times, washed with methionine-free medium, and pulse-labeled for 20 min in the methionine-free medium with 10 $\mu$Ci [$^{35}$S]-methionine. The cells were then washed with ice-cold PBS, and proteins were extracted in lysis buffer. Equal amount of proteins were precipitated with 7.5% trichloroacetic acid, and radioactivity was measured by liquid scintillation counting.

### Metabolic labeling with AHA and proteomic analysis

To estimate the effect of TG2 inhibition in hypoxia, A549 cells with or without 500 $\mu$M cystamine in hypoxia (1% O$_2$) were incubated with L-methionine–free RPMI media with 10% dialyzed FBS for 30 min to enable AHA uptake and then labeled with 50 $\mu$M AHA (Click Chemistry Tools) for 18 h. AHA-labeled proteins in the extracts were purified using Click-&-Go Protein Enrichment Kit (Click Chemistry Tools) according to the manufacturer's protocols. AHA-conjugated proteins were enriched by incubation with alkyne-agarose resin followed by rigorous washing with agarose SDS wash buffer (100 mM Tris–Cl, pH 8.0, 1% SDS, 250 mM NaCl, and 5 mM EDTA).

Peptide separation was performed using Dionex UltiMate 3000 RSLCnano system (Thermo Fisher Scientific). Tryptic peptides from bead column were reconstituted using 0.1% formic acid and separated on a 50-cm Easy-Spray column with a 75-$\mu$m inner diameter packed with 2 $\mu$m C18 resin (Thermo Fisher Scientific) over 120 min (300 nl/min) using a 0–45% acetonitrile gradient in 0.1% formic acid at 50°C. The LC was coupled to a Q Exactive mass spectrometer with a nano-ESI source. Mass spectra were acquired in a data-dependent mode with an automatic switch between a full scan with five data-dependent tandem mass spectrometry (MS/MS) scans. Target value for the full scan MS spectra was 3,000,000 with a maximum injection time of 120 ms and a resolution of 70,000 at m/z 400. The ion target value for MS/MS was set to 1,000,000 with a maximum injection time of 120 ms and a resolution of 17,500 at m/z 400. Dynamic exclusion of repeated peptides was applied for 20 s.

Resulting raw files were processed using Proteome Discoverer (version 2.2; Thermo Fisher Scientific) for identification with the database of *Homo sapiens* (organism ID: 9606, 71567 entries, UniProt). The search parameters were set as default, including cysteine carbamidomethylation as a fixed modification, and N-terminal acetylation and methionine oxidation as variable modifications with two miscleavages. Peptide identification was based on a search with an initial mass deviation of the precursor ion of up to 10 ppm, and the allowed fragment mass deviation was set to 20 ppm. Label-free quantitation was performed using peak intensity for unique and razor peptide of each protein. Normalization was done using total peptide amount.

### 5′UTR analysis

5′UTRs of the 342 genes identified only in hypoxia condition and 96 genes identified only in hypoxia + cystamine condition were obtained from UTRdb (http://utrdb.ba.itb.cnr.it/). The length and GC contents were calculated using DNA/RNA GC Content Calculator (http://www.endmemo.com/bio/gc.php). The ΔG values were estimated using Quikfold from The DINAMelt Web Server (http://unafold.rna.albany.edu/?q=DINAMelt/Quickfold). The UTRdb and Database of Transcriptional Start Sites (DBTSS Release 10.1; https://dbtss.hgc.jp/) were used to identify putative transcription start sites and genes with putative uORF and 5′ TOP sequences in 5′UTR.

### Statistical analysis

Statistical calculations were performed using Prism 4.0 (GraphPad). Differences between two variables were assessed by two-sided *t* test for continuous variables or chi-square test for categorical variables. The difference was considered significant if the *P*-value was less than 0.05.

# Supplementary Information

# Acknowledgements

This work was supported by the National Research Foundation of Korea (NRF) grant funded by the Korea government (Ministry of Science & ICT [Information & Communication Technology]) (grants 2012R1A1A2005188, 2017R1C1B2002183, 2017M3A9B4061890, and NRF-2018R1A2B3008541); the Bio & Medical Technology Development Program of the NRF funded by the Ministry of Science & ICT (grant no. 2018M3A9F3056902); the Korea Healthcare Technology R&D Project, Ministry of Health and Welfare, Republic of Korea (HI18C2396); and the Brain Korea 21 PLUS program of the Korean Ministry of Education, Science and Technology.

## Author Contributions

S-Y Cho: conceptualization, data curation, formal analysis, supervision, funding acquisition, validation, investigation, and writing—original draft, review, and editing.
S Lee: formal analysis, validation, and investigation.
J Yeom: formal analysis and investigation.
H-J Kim: formal analysis, validation, and investigation.
J-H Lee: formal analysis, validation, and investigation.
J-W Shin: investigation.
M-a Kwon: investigation.
KB Lee: investigation.
EM Jeong: investigation.
HS Ahn: formal analysis and investigation.
D-M Shin: conceptualization.
K Kim: conceptualization, data curation, formal analysis, validation, and investigation.
I-G Kim: conceptualization, supervision, funding acquisition, investigation, and writing—original draft, review, and editing.

## Conflict of Interest Statement

The authors declare that they have no conflict of interest.

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
