## [Reviewer comments · Life Science Alliance]

Life Science Alliance

Transglutaminase 2 mediates hypoxia-induced selective mRNA translation via polyamination of 4EBPs

Sung-Yup Cho, Seungun Lee, Jeonghun Yeom, Hyo-Jun Kim, Jin-Haeng Lee, Ji-Woong Shin, Mee-ae Kwon, Ki Lee, Eui Jeong, Hee Ahn, Dong-Myung Shin, Kyunggon Kim, and In-Gyu Kim

DOI: <https://doi.org/10.26508/lsa.201900565>

Corresponding author(s): Sung-Yup Cho, Seoul National University College of Medicine; In-Gyu Kim, Seoul National University College of Medicine; and Kyunggon Kim, Seoul National University

Review Timeline:	Submission Date:	2019-09-29
	Editorial Decision:	2019-10-15
	Revision Received:	2020-01-06
	Editorial Decision:	2020-01-21
	Revision Received:	2020-02-05
	Accepted:	2020-02-06

Scientific Editor: Andrea Leibfried

Transaction Report:

October 15, 2019

Re: Life Science Alliance manuscript #LSA-2019-00565-T

Sung-Yup Cho
Seoul National University College of Medicine

Dear Dr. Cho,

Thank you for submitting your manuscript entitled "Transglutaminase 2 mediates hypoxia-induced selective mRNA translation via polyamination of 4EBPs" to Life Science Alliance. The manuscript was assessed by expert reviewers, whose comments are appended to this letter.

As you will see, the reviewers appreciate your data. However, they think that some of your conclusions are not sufficiently supported by the data provided. We would like to invite you to submit a revised version of your work, addressing the individual criticisms raised. Though more support for your conclusions and quite some additional controls are needed, the revision seems rather straightforward to perform. However, please do get in touch in case you would like to discuss individual revision points further.

Thank you for this interesting contribution to Life Science Alliance. We are looking forward to receiving your revised manuscript.

Sincerely,

B. MANUSCRIPT ORGANIZATION AND FORMATTING:

Reviewer #1 (Comments to the Authors (Required)):

Sung-Yup Cho et al. showed that under hypoxia condition, Transglutaminase 2(TG2) promotes eIF4E associated 4E-BP1's polyamination. Polyaminated 4E-BP1 has an increased binding affinity with Raptor, leading to its phosphorylation by mTORC1, thus promoting the translation of a subset of mRNAs with GC-rich 5'UTR.

In general, it is an interesting and novel mechanism showing that TG2 elicits 4E-BPs polyamination in eIF4E dependent manner and promotes 4E-BP1 inactivation by mTORC1 under hypoxia condition. However, some issues and concerns must be addressed.

Major concerns:

1. The authors only showed that polyaminated 4E-BP1 associated more strongly with Raptor in vitro (Fig. 4D). However, under hypoxia condition, mTORC1 is inactivated, thus, their results would be more convincing if they can show that purified Raptor-mTOR complex from hypoxic cells is able to phosphorylate polyaminated 4E-BP1 in vitro.
2. The authors showed that TG2 regulates eIF4E dependent mRNA translation by Cap binding assay (Fig. 4B). However, the m7GTP pull-down experiment only demonstrated a small difference between control and TG2 knockdown cells, which have been exposed to hypoxia for 48 hours (Fig. 4B). Since they showed that the TG2 is required for maintaining 4E-BP1 phosphorylation in cells, which have been exposed to hypoxia for 24 hours (Fig. 4A), they should use these cells to repeat their experiment. Moreover, by using a bicistronic translation reporter, the authors showed that overexpression of TG2 promoted cap-dependent translation under normoxic conditions in polyaminated 4E-BP1-dependent manner (Fig. 5A, B and C). However, it should be noted that TG2 overexpression may promote PI3K/mTOR activity (Boroughs LK et al., 2014) and downregulate PTEN activity (Wang Y et al., 2014). The authors, therefore, should examine the activities of these regulators under TG2 overexpression.
3. Generally, many figures, the authors did not show all the controls necessary for supporting their results. Some examples; Fig. 1C lacks the TG2 immunoblotting (IB), Fig. 3C lacking INPUT; Fig. 5, the IB lacks the expression of TG2 and 4E-BP1.

Minor points:

1. Fig. 1A, the authors should indicate that pull-downed 4E-BP1 corresponds to the upper or lower band of 4E-BP1 in the input.
2. Fig. 1G, why in PP242 treated cells there is not a clearly increased polyaminated 4E-BP1?
3. Fig. 1H, why Myc tagged 4E-BP1 appears only as one band? Phospho-4E-BP1 antibody should be used to evaluate the phosphorylation state of 4E-BP1-MYC.
4. Fig. 3B, in eIF4E knockdown cells, 4E-BP1 seems to be more phosphorylated as there are clearly increased upper bands. According to the authors, phosphorylated 4E-BP1 will be less polyaminated. The authors thus should use the phospho-4E-BP1 antibody to examine the 4E-BP1 phosphorylation state in eIF4E knockdown cells.
5. Fig. 4C, the authors should explain why AKT and S6K dramatically decreased in cells that have been exposed to hypoxia for 24 and 48 hours?
6. Fig. 4D, why Spermine alone without TG2 can also clearly increase 4E-BP1 and raptor binding?

A novel mechanism by which tissue transglutaminase activates signaling events that promote cell survival. Boroughs LK, Antonyak MA, Cerione RA. *J Biol Chem*. 2014 Apr 4; 289(14):10115-25.

Phosphorylation of transglutaminase 2 (TG2) at serine-216 has a role in TG2 mediated activation of nuclear factor-kappa B and in the downregulation of PTEN. Wang Y, Ande SR, Mishra S. *BMC Cancer*. 2012 Jul 3; 12():277.

Reviewer #2 (Comments to the Authors (Required)):

The manuscript from Cho et al describes work to address the role of TG2 in polyamination of 4E-BP1 in response to hypoxia. Previous work has shown that 4E-BP1 is subject to phosphorylation, ubiquitination and cleavage as modes of regulating its activity. The work here shows that 4E-BP1 is also regulated by polyamination which enhances 4E-BP1 affinity for raptor, thereby enhancing phosphorylation by mTORC1. Furthermore, it is shown that translation of a subset of mRNAs, ric in GC content, is preferentially enhanced by polyamination of 4E-BP1.

As such, the conclusions are supported by the data shown. For completeness, in my opinion, the authors need to address the following points:

1. p4; removed reference to 4E-BP3 here as it is not relevant to the argument made here and it not addressed during this study.
 2. Fig 1; could the authors mentions that this is a high Km and comment of the physiological relevance.
 3. Fig.4C; there is a decrease in total Akt and p70S6K at 24hrs with shTG2; this needs to be addressed in the text and the reasons discussed. What happens with Akt-S308 phosphorylation which is more of an indicator of PI3-K activity? The Western blot for Raptor is unclear; a better version needs to be presented.
 4. Fig.4D; why use HeLa extracts here? What happens with Q28/93/113A here?
 5. Fig.5A; what happens with CTA alone in this assay? Does CTA decreased TG2 expression?
 6. Fig.5C; The authors need to provide a Western blot showing equal levels of protein expression between variables.
- Fig.5F; why use MCF7 cells here? What does the scale mean? The authors need to provide a Western blot showing levels of protein expression from 3 independent experiments with appropriate statistical analysis.

Please find below our responses to each specific comment and suggestion raised by the two reviewers.

Reviewer #1:

1. The authors only showed that polyaminated 4E-BP1 associated more strongly with Raptor in vitro (Fig. 4D). However, under hypoxia condition, mTORC1 is inactivated, thus, their results would be more convincing if they can show that purified Raptor-mTOR complex from hypoxic cells is able to phosphorylate polyaminated 4E-BP1 in vitro.

: To check the kinase activity of mTORC1 in hypoxic condition, we examined the phosphorylation status of purified 4EBP1 from bacteria (4EBP1-GST) after incubating with cell lysate of hypoxic cells. When blotted with p-4EBP1 antibody, purified 4EBP1 was phosphorylated by cell lysate of hypoxic cells (1% O₂ for 6 h; Fig. 4E). Moreover, the polyamination of 4EBP1 by TG2 increased the phosphorylation levels of purified 4EBP1 (Fig. 4E), suggesting that increased binding between polyaminated 4EBP1 and Raptor enhanced the mTORC1-mediated 4EBP1 phosphorylation. These results are now included in the Results sections of the revised manuscript (p. 10) and in Figure 4E.

2. The authors showed that TG2 regulates eIF4E dependent mRNA translation by Cap binding assay (Fig. 4B). However, the m⁷GTP pull-down experiment only demonstrated a small difference between control and TG2 knockdown cells, which have been exposed to hypoxia for 48 hours (Fig. 4B). Since they showed that the TG2 is required for maintaining 4E-BP1 phosphorylation in cells, which have been exposed to hypoxia for 24 hours (Fig. 4A), they should use these cells to repeat their experiment.

: We performed the m⁷GTP pull-down experiment for cells which have been exposed to hypoxia for 24 hours as reviewer suggested. We found that knock-down of TG2 significantly increased the binding of 4EBP1 to eIF4E in hypoxic condition for 24 hours (Fig. 4B). These results are now included in the Results sections of the revised manuscript (p. 9) and in Figure 4B, and the data from cells, which have been exposed to hypoxia for 48 hours, was demonstrated in Supplemental Fig. S3.

3. Moreover, by using a bicistronic translation reporter, the authors showed that overexpression of TG2 promoted cap-dependent translation under normoxic conditions in polyaminated 4E-

BP1-dependent manner (Fig. 5A, B and C). However, it should be noted that TG2 overexpression may promote PI3K/mTOR activity (Boroughs LK et al., 2014) and downregulate PTEN activity (Wang Y et al., 2014). The authors, therefore, should examine the activities of these regulators under TG2 overexpression.

: When we overexpressed TG2 in SK-N-SH cells and examined the phosphorylation status of AKT and S6K, TG2 overexpression had little effect on the phosphorylation status of AKT and S6K (Supplemental Fig. S6A), indicating that the intrinsic activity of AKT and mTORC1 hardly changed by TG2 expression levels in this cell line. In addition, the expression levels of PTEN did not change by TG2 overexpression (Supplemental Fig. S6A). These results are now included in the Results sections of the revised manuscript (p. 11) and in Supplemental Fig. S6A.

4. Generally, many figures, the authors did not show all the controls necessary for supporting their results. Some examples; Fig. 1C lacks the TG2 immunoblotting (IB), Fig. 3C lacking INPUT; Fig. 5, the IB lacks the expression of TG2 and 4E-BP1.

: As the reviewer suggested, we added the data of control immunoblotting in Fig. 1C (For TG2), Fig. 3C (for eIF4E and TG2), Fig. 3D (for TG2), and Fig. 5 (for TG2 and 4EBP1).

5. Fig. 1A, the authors should indicate that pull-downed 4E-BP1 corresponds to the upper or lower band of 4E-BP1 in the input.

: The pull-downed 4E-BP1 corresponds to lower band of 4E-BP1 in the input. These results are now included in the Results sections of the revised manuscript (p. 6) and legend of Fig. 1A.

6. Fig. 1G, why in PP242 treated cells there is not a clearly increased polyaminated 4E-BP1?

: We replaced the figure with clearer blot, which shows increased polyaminated 4E-BP1 (Fig. 1G).

7. Fig. 1H, why Myc tagged 4E-BP1 appears only as one band? Phospho-4E-BP1 antibody should be used to evaluate the phosphorylation state of 4E-BP1-MYC.

: When we performed the experiment again, we found that the migration of wild-type 4EBP1-myc was a little slower than T37,46A-mutant 4EBP1-myc (Fig. 1H, input), suggesting that the phosphorylation status of wild-type 4EBP1-myc was higher than that of mutant 4EBP1-myc. Consistent to this result, the phosphorylation of the Myc-immunoprecipitated 4EBP1 was

detected only in wild-type 4EBP1 (Fig. 1H; Myc-IP). These results are now included in Fig. 1H and legend of Fig. 1H.

8. Fig. 3B, in eIF4E knockdown cells, 4E-BP1 seems to be more phosphorylated as there are clearly increased upper bands. According to the authors, phosphorylated 4E-BP1 will be less polyaminated. The authors thus should use the phospho-4E-BP1 antibody to examine the 4E-BP1 phosphorylation state in eIF4E knockdown cells.

: As the reviewer suggested, knock-down of eIF4E slightly increased the phosphorylated 4E-BP1 (Fig. 3B). This effect was probably due to the feedback effect to increase the cap-dependent translation, because the knockdown of eIF4E readily inhibited the cap-dependent translation. However, the decrease of 4EBP1 polyamination was significant compared to the increase of 4EBP1 phosphorylation (Fig. 3B). Therefore, we suggest that eIF4E is required for the polyamination of 4EBP1. These results are now included in the Results sections of the revised manuscript (p. 8) and Fig. 3B.

9. Fig. 4C, the authors should explain why AKT and S6K dramatically decreased in cells that have been exposed to hypoxia for 24 and 48 hours?

: When we performed the experiment again, we found that the total AKT and S6K levels slightly decrease in hypoxia differently with our previous results, and these results have been repeatedly verified. These difference probably are dependent on the cell culture condition and status of stably transfected cells. Previous reports showed that AKT and S6K were substrates of caspase 3 (Rokudai et al., *J Cell Physiol.* 2000;182(2):290-6, Cho et al., *Mol Cancer Ther.* 2017;16(10):2178-2190), and hypoxia is one of stresses causing caspase 3 activation (Malhotra et al., *Am J Physiol Cell Physiol.* 2001;281(5):C1596-603). Therefore, AKT and S6K are possibly cleaved by caspase activation in hypoxic conditions, but it seems to be context dependent. We replaced the blots for AKT and S6K in Fig 4C with the more reproducible results in our experimental setting. And the explanation of the decrease of total AKT and S6K levels in hypoxia was included in the Discussion sections of the revised manuscript (p. 16).

10. Fig. 4D, why Spermine alone without TG2 can also clearly increase 4E-BP1 and raptor binding?

: Polyamines are positively charged small molecules and previously reported to be able to modulate the electrostatic protein-protein interactions (Berwanger et al., *J Inorg Biochem.*

2010;104(2):118-25, Thomas et al., J Mol Endocrinol. 1999;22(2):131-9). In our experiment, spermine increased the interaction between 4EBP1 and Raptor. However, the exact molecular mechanisms of enhancing the interaction by spermine need to be further studied. And, incorporation of spermine to 4EBP1 by TG2 still enhanced the interaction between 4EBP1 and Raptor (Fig. 4E and Supplemental Fig. S4). These are now included in the Discussion sections of the revised manuscript (p. 16).

Reviewer #2:

1. p4; removed reference to 4E-BP3 here as it is not relevant to the argument made here and it not addressed during this study.

: We showed that 4EBP1, -2, -3 are substrates of TG2 and identified the target glutamine site for polyamination (Fig. 2). We investigated the functional change of TG2-mediated modification using 4EBP1.

2. Fig 1; could the authors mentions that this is a high Km and comment of the physiological relevance.

: Although the Km values for TG2-mediated polyamination of 4EBPs were high, concentration of intracellular polyamines was near 1 mM (Igarashi & Kashiwagi, Int J Biochem Cell Biol. 2010;42(1):39-51). Although most of them were bound to nucleic acid, protein, and phospholipid, the concentration of free polyamines was about 20 – 200 μ M. Therefore, we suggest that 4EBPs can be modified by TG2 in cellular context. These are now included in the Discussion sections of the revised manuscript (p. 16).

3. Fig.4C; there is a decrease in total Akt and p70S6K at 24hrs with shTG2; this needs to be addressed in the text and the reasons discussed. What happens with Akt-S308 phosphorylation which is more of an indicator of PI3-K activity? The Western blot for Raptor is unclear; a better version needs to be presented.

: When we performed the experiment again, we found that the total AKT and S6K levels slightly decrease in hypoxia differently with our previous results, and these results have been repeatedly verified. These difference probably are dependent on the cell culture condition and status of stably transfected cells. Previous reports showed that AKT and S6K were substrates of caspase 3 (Rokudai et al., J Cell Physiol. 2000;182(2):290-6, Cho et al., Mol Cancer Ther. 2017;16(10):2178-2190), and hypoxia is one of stresses causing caspase 3 activation (Malhotra

et al., *Am J Physiol Cell Physiol.* 2001;281(5):C1596-603). Therefore, AKT and S6K are possibly cleaved by caspase activation in hypoxic conditions resulting in the decreased expression levels, but it seems to be context dependent and need to be further studied. The explanation of the decrease of total AKT and S6K levels in hypoxia was included in the Discussion sections of the revised manuscript (p. 16). We also checked Akt-S308 phosphorylation and found little significant differences between wild-type and TG2-knockdown cells (Fig. 4C). These are now included in the Result sections of the revised manuscript (p. 10) and Fig. 4C. We replaced the blots for AKT, S6K, mTOR and Raptor in Fig 4C with the more reproducible results in our experimental setting.

4. Fig.4D; why use HeLa extracts here? What happens with Q28/93/113A here?

: We performed the experiment again using hypoxic A549 cell lysate, and showed similar results (Fig. 4E). For consistence of manuscript, we replaced the Fig. 4E with results from A549 cells, and moved the data from HeLa cells to Supplemental Fig. S4. When we adopted the 4EBP1 Q28/93/113A mutant in this experimental setting, we found that modification of mutant by TG2 did not increase the interaction between 4EBP1 and Raptor (Fig. 4F). These results are now included in the Results sections of the revised manuscript (p. 11) and in Fig. 4D-F.

5. Fig.5A; what happens with CTA alone in this assay? Does CTA decreased TG2 expression?

: CTA treatment slightly increased the IRES translation activity, but little effect on cap-dependent translation in SK-N-SH cells (Supplemental Fig. S6B). Because SK-N-SH cells did not express TG2 (Supplemental Fig. S6A), this effect is not associated with the change of TG2 expression. These results are now included in the Results sections of the revised manuscript (p. 11) and in Supplemental Fig. S6B.

6. Fig.5C; The authors need to provide a Western blot showing equal levels of protein expression between variables.

: We added the control western blots for overexpression of TG2 and 4EBP1 in Fig. 5.

7. Fig.5F; why use MCF7 cells here? What does the scale mean? The authors need to provide a Western blot showing levels of protein expression from 3 independent experiments with appropriate statistical analysis.

: In Fig. 5F, we adopted HEK293 cells stably transfected with control and TG2-overexpressing vectors (Cho et al., *Exp Mol Med.* 2010;42(9):639-50). We adopted these cell lines to assay the protein synthesis in stably TG2-overexpressing system. We added the western blot showing the overexpression of TG2 in HEK293 cells (Fig. 5F). The incorporation of S35-methionine was depicted compared to control HEK293 cells at 0 h. We clearly indicated the used cell lines and the scale in the legend of Fig. 5F.

January 21, 2020

RE: Life Science Alliance Manuscript #LSA-2019-00565-TR

Prof. Sung-Yup Cho
Seoul National University College of Medicine
103 Daehak-ro
Seoul 03080
Korea (South), Republic of

Dear Dr. Cho,

Thank you for submitting your revised manuscript entitled "Transglutaminase 2 mediates hypoxia-induced selective mRNA translation via polyamination of 4EBPs". As you will see, the reviewers largely appreciate the changes introduced in revision, and we would thus be happy to publish your paper in Life Science Alliance pending final minor revisions:

- Please address reviewer #2's remaining concerns
- Please upload the suppl figures as individual files; the legends should get incorporated into the main manuscript docx file
- Please add a callout in the manuscript text to Figure 1B
- Please upload the supplementary tables in either word docx or excel format and provide legends for them (the latter can go into the main manuscript file)
- Please mention the statistical test performed next to the p-values mentioned in the figure legends
- Please mention throughout your manuscript (eg in figure legends) the number of replicates for the data provided

A. FINAL FILES:

-- High-resolution figure, supplementary figure and video files uploaded as individual files: See our

detailed guidelines for preparing your production-ready images, <http://www.life-science-alliance.org/authors>

B. MANUSCRIPT ORGANIZATION AND FORMATTING:

Sincerely,

Andrea Leibfried, PhD
Executive Editor
Life Science Alliance
Meyerhofstr. 1
69117 Heidelberg, Germany
t +49 6221 8891 502

Reviewer #1 (Comments to the Authors (Required)):

The authors have addressed all of the concerns of this referee in full. The manuscript, in its revised form, is more compelling and complete.

Reviewer #2 (Comments to the Authors (Required)):

The authors addressed the previous concerns satisfactorily, but they need to answer to a few minor concerns.

1. The authors must indicate when cells were exposed to hypoxia (for example, Fig 1A, G, H, ...).
2. 'Pull-down' and 'Strep-HRP' are interchangeably used in the figures. The authors should use one of them.
3. Fig 4F. It looks like spermidine alone increases 4E-BP phosphorylation and its binding to Raptor without TG2. The authors need to show 4E-BP1 WT alone as a control.
4. Fig 5. The effect of TG2 on cap-dependent and cap-independent translation is inconsistent (for example, IRES activity by TG2 in Fig 5A and Fig 5B).

Responses to Reviewer's comments:

Please find below our responses to each specific comment and suggestion raised by the editor and reviewers.

Editor:

1. Please address reviewer #2's remaining concerns.

: We addressed the reviewer #2's concerns one by one in latter part of 'Responses to Reviewer's comments'.

2. Please upload the suppl figures as individual files; the legends should get incorporated into the main manuscript docx file.

: We prepared the supple figures as individual TIFF files and the legends were incorporated into the main manuscript docx file.

3. Please add a callout in the manuscript text to Figure 1B.

: We added a callout in the manuscript text to Figure 1B in page 6.

4. Please upload the supplementary tables in either word docx or excel format and provide legends for them (the latter can go into the main manuscript file).

: We prepared the supplementary tables in excel format and provide legends for them in the main manuscript file.

5. Please mention the statistical test performed next to the p-values mentioned in the figure legends.

: We addressed the statistical test performed in the legends of Fig. 6B, 6C, 6D, 7A, and 7B.

6. Please mention throughout your manuscript (eg in figure legends) the number of replicates for the data provided.

: We addressed the number of replicates for the data provided in the legends of Fig. 1C, 2A, 2C, 5A-F, and 7C.

Reviewer #2:

1. The authors must indicate when cells were exposed to hypoxia (for example, Fig 1A, G,

H, ...).

: We indicated more clearly when cells were exposed to hypoxia in the legends of Fig. 1A, 1G, 1H, 3A, 3B, 3I, 4B, 4E, and 4F.

2. 'Pull-down' and 'Strep-HRP' are interchangeably used in the figures. The authors should use one of them.

: We used 'Pull-down' and 'Strep-HRP' in different ways. We used 'Pull-down' when the BP-incorporated proteins were purified by streptavidin pull-down and detected by western blot analysis with anti-bodies of BP-incorporated proteins (for example, 4EBP1). But we used 'Strep-HRP' when the BPs were incorporated into purified proteins and detected by western blot analysis using streptavidin-HRP for development. To avoid the confusion, we changed 'BP incorporated into 4EBP1 was detected' into 'BP-incorporated 4EBP1 was detected' in case of 'Pull-down' in Fig. 1A, 1F-H, and 3A.

3. Fig 4F. It looks like spermidine alone increases 4E-BP phosphorylation and its binding to Raptor without TG2. The authors need to show 4E-BP1 WT alone as a control.

: We added the blots for 4E-BP1 WT as a control in Fig. 4F. As reviewer indicated, spermidine alone increased the interaction between 4EBP1 and Raptor, and we already discussed the electrostatic effect of positively charged polyamines in the Discussion session (p. 16).

4. Fig 5. The effect of TG2 on cap-dependent and cap-independent translation is inconsistent (for example, IRES activity by TG2 in Fig 5A and Fig 5B).

: As the reviewer indicated, the effect of TG2 on IRES-dependent translation, which were estimated by luciferase activity, was inconsistent probably according to cell culture conditions. However, the effect of TG2 on cap-dependent translation, which was estimated by CAT activity, was consistent, because the overexpression of TG2 consistently increased the CAT activity (Fig. 5A and 5B). In this study, we focused on the effect of TG2 on cap-dependent translation, and the effect of TG2 on IRES-dependent translation need further studies.

February 6, 2020

RE: Life Science Alliance Manuscript #LSA-2019-00565-TRR

Prof. Sung-Yup Cho
Seoul National University College of Medicine
103 Daehak-ro
Seoul 03080
Korea (South), Republic of

Dear Dr. Cho,

Thank you for submitting your Research Article entitled "Transglutaminase 2 mediates hypoxia-induced selective mRNA translation via polyamination of 4EBPs". I appreciate the introduced changes and it is a pleasure to let you know that your manuscript is now accepted for publication in Life Science Alliance. Congratulations on this interesting work.

DISTRIBUTION OF MATERIALS:

Again, congratulations on a very nice paper. I hope you found the review process to be constructive and are pleased with how the manuscript was handled editorially. We look forward to future exciting submissions from your lab.

Sincerely,
